# Motor cortex signals for each arm are mixed across hemispheres and neurons yet partitioned within the population response

Katherine Cora Ames[1,2,3,4]*, Mark M Churchland[1,2,3,5]

[1]Department of Neuroscience, Columbia University, New York, United States; [2]Zuckerman Institute, Columbia University, New York, United States; [3]Grossman Center for the Statistics of Mind, Columbia University, New York, United States; [4]Center for Theoretical Neuroscience, Columbia University, New York, United States; [5]Kavli Institute for Brain Science, Columbia University, New York, United States

**Abstract** Motor cortex (M1) has lateralized outputs, yet neurons can be active during movements of either arm. What is the nature and role of activity across the two hemispheres? We recorded muscles and neurons bilaterally while monkeys cycled with each arm. Most neurons were active during movement of either arm. Responses were strongly arm-dependent, raising two possibilities. First, population-level signals might differ depending on the arm used. Second, the same population-level signals might be present, but distributed differently across neurons. The data supported this second hypothesis. Muscle activity was accurately predicted by activity in either the ipsilateral or contralateral hemisphere. More generally, we failed to find signals unique to the contralateral hemisphere. Yet if signals are shared across hemispheres, how do they avoid impacting the wrong arm? We found that activity related to each arm occupies a distinct subspace, enabling muscle-activity decoders to naturally ignore signals related to the other arm.
DOI: https://doi.org/10.7554/eLife.46159.001

*For correspondence:
kca2120@columbia.edu

Competing interests: The authors declare that no competing interests exist.

## Introduction

The outputs of motor cortex (M1) are lateralized: most spinal projections influence the contralateral musculature. M1 lesions thus produce contralateral motor deficits (*Liu and Rouiller, 1999*; *Murata et al., 2008*; *Passingham et al., 1983*; *Vilensky and Gilman, 2002*). Similarly, electrical microstimulation activates contralateral musculature (*Kwan et al., 1978*; *Sessle and Wiesendanger, 1982*). The degree to which computations within M1 are lateralized remains less clear. The corpus callosum connects M1 across hemispheres, yielding the potential for extensive cooperation (*Gould et al., 1986*; *Jenny, 1979*; *Jones and Wise, 1977*). Callosally mediated interactions are readily revealed by paired-pulse TMS protocols and can involve net facilitation or suppression (*Ferbert et al., 1992*; *Hanajima et al., 2001*; *Meyer et al., 1995*). An obvious role for inter-hemispheric communication is coordination of bimanual movement (*Donchin et al., 1998*; *Haken et al., 1985*; *Kelso, 1984*; *Kermadi et al., 1998*). Yet, there is evidence that unimanual movements also involve sharing information across hemispheres.

Most physiological studies of unimanual movements have focused on activity contralateral to the moving limb, on the grounds that contralateral activity is most functionally relevant and likely to be most prevalent. Yet, ipsilateral activity can be robust. During finger movements, ipsilateral single-neuron activity is modest but present (*Aizawa et al., 1990*; *Matsunami and Hamada, 1981*;

*Tanji et al., 1988*) and behavior can be decoded from BOLD activation within ipsilateral motor cortex (*Berlot et al., 2019*; *Diedrichsen et al., 2013*). Considerable ipsilateral activity has been reported during movements involving the upper arm, such as reaching to remove food from a drawer (*Kermadi et al., 1998*; *Kazennikov et al., 1999*), or performing center-out reaches (*Cisek et al., 2003*; *Donchin et al., 2002*; *Ganguly et al., 2009*; *Steinberg et al., 2002*).

While the presence of ipsilateral activity is established, the nature of that activity is less clear. Few studies have directly compared neural response patterns when the same movement is performed by one arm versus the other. In premotor areas, delay-period responses can encode information about an upcoming reach (*Cisek et al., 2003*) or grasp (*Michaels and Scherberger, 2018*) independently of which arm would subsequently move, suggesting that preparatory activity is largely limb-independent. However, activity during movement was more limb-dependent, for both premotor cortex and M1 (*Cisek et al., 2003*). *Steinberg et al. (2002)* reported similar single-neuron directional tuning in M1 regardless of which arm was moving, yet also found evidence for limb-dependent population-level encoding of direction. *Donchin et al. (1998)* found that activity during bimanual movements differed from that during unimanual movements, but did not analyze whether response tuning varied between contralateral and ipsilateral movements. Thus, it remains unclear to what degree the pattern of M1 responses – at either the single-neuron or population levels – depends on the limb being used.

If responses are limb-independent then the relationship between hemispheres is necessarily simple: both contain the same information, encoded in the same manner. In contrast, strongly limb-dependent responses would raise additional questions. Are 'lower-level' signals (e.g. those describing muscle activity) more prevalent in the contralateral hemisphere? More generally, which signals are shared across hemispheres? How is activity structured such that only one arm moves when both hemispheres are active?

We investigated these questions using a novel 'cycling' task, performed with either the left or right arm. We recorded neural activity from both hemispheres simultaneously. In separate sessions, we recorded muscle activity bilaterally. Individual neurons responded robustly regardless of which arm performed the task. Yet, responses were strongly limb-dependent: for a given neuron, response patterns pertaining to the two arms were essentially unrelated. Nevertheless, we found no clear evidence that the two hemispheres contained different types of information. For example, muscle activity could be decoded equally well from contralateral or ipsilateral neural activity. More broadly, any signal that was strongly present in one hemisphere was present (and similarly strong) in the other. Thus, activity in a given hemisphere contains similar information during movement of one arm versus the other, but that information is distributed very differently across individual neurons. This might appear to yield a paradox: how can M1 be robustly active without driving the contralateral arm? A solution emerged when we examined population activity: arm-specific signals were partitioned into orthogonal dimensions, allowing a simple decoder to naturally separate signals related to the two arms.

## Results

### Terminology

We adopt the following terminology. For neurons in a given hemisphere, we refer to the contralateral arm as the 'driven arm' (reflecting the strong connections to the contralateral spinal cord). We refer to the ipsilateral arm as the 'non-driven arm.' Thus, for a neuron recorded from the right hemisphere, the left arm is the driven arm and the right arm is the non-driven arm. For the muscles, the driven arm is simply the arm upon which the muscle acts. We use the term 'driving cortex' to refer to the hemisphere contralateral to the performing arm. We use the term 'non-driving cortex' to refer to the hemisphere ipsilateral to the performing arm.

### Behavior

Two monkeys (E and F) were trained on a cycling task that could be performed with either arm (*Figure 1A*). Left and right hands each grasped a pedal. Monkeys performed blocks of left-hand and right-hand trials. Cycling the correct pedal produced motion through the virtual environment. Success required that the non-performing arm be kept still. On each trial, monkeys cycled from one

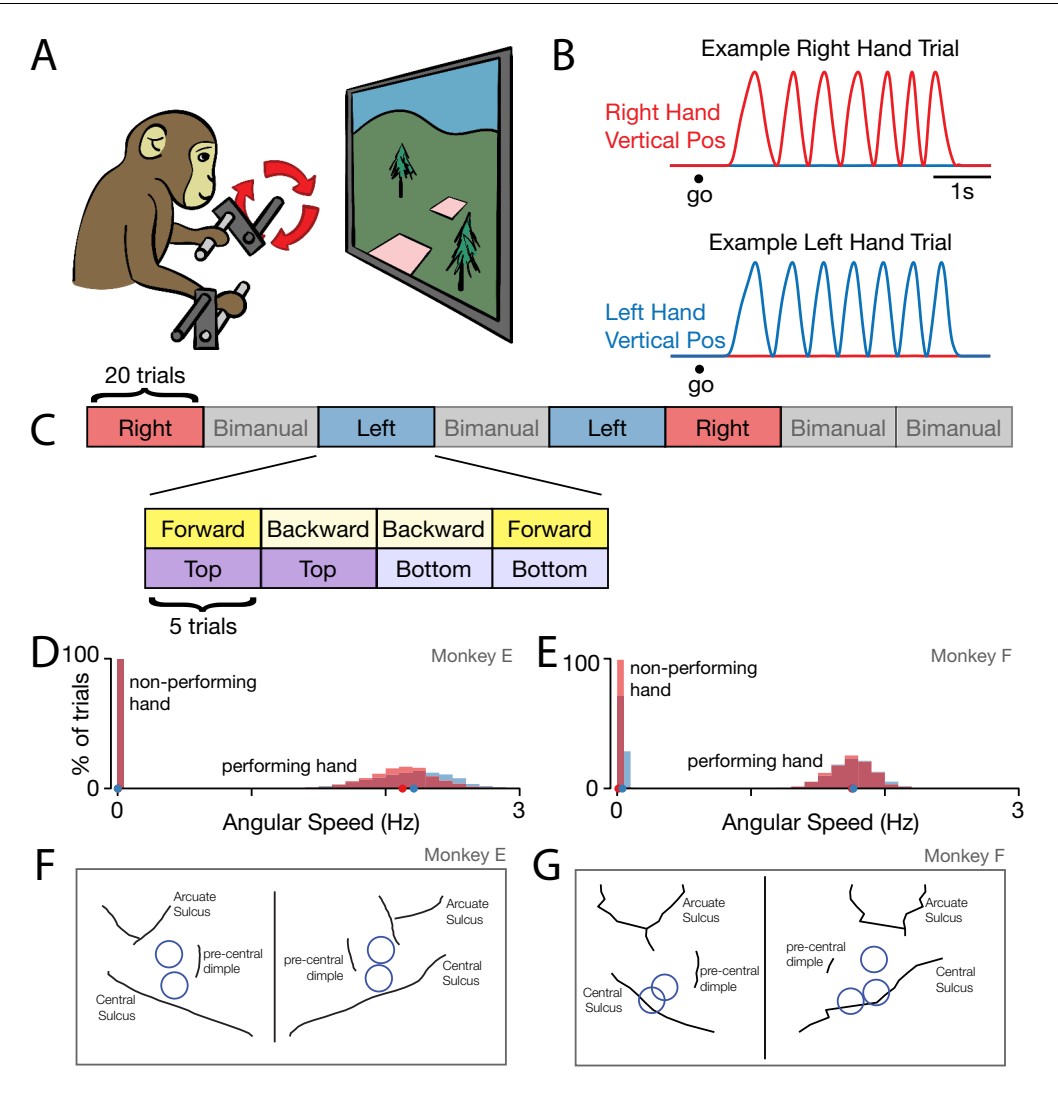

**Figure 1.** Behavior. (A) Task schematic. Cycling one of the two pedals produced progress through a virtual environment. The other pedal had to remain stationary. This schematic simplifies the physical setup. In particular, pedals employed a handle that ensured consistent hand posture and a brace that minimized wrist movement. (B) Behavior on two example trials. After a go cue, the monkey cycled for seven cycles with one hand while holding the other hand stationary. Red trace: Right hand vertical position. Blue trace: Left hand vertical position. (C) Task structure. Blocks of right hand, left hand, and bimanual conditions were presented in pseudorandom order. Within each block of 20 trials, trials were presented in sub-blocks of five trials. There was one sub-block for each combination of cycling direction and starting position. (D) Distributions of cycling speed for the performing and non-performing hands. Distributions are across trials. Within each trial, the average angular speed was computed after taking the magnitude of velocity at each time. Red: Right hand. Blue: Left hand. Dots show distribution medians. Data are for Monkey E. (E) Same as D but for Monkey F. (F) For Monkey E, location of burr holes from which recordings were made, superimposed on MR-derived anatomical map. (G) Same as F, for Monkey F.
DOI: https://doi.org/10.7554/eLife.46159.002

target to another, located seven cycles away (*Figure 1B*). Targets were positioned so that cycling started and ended either at the top of the cycle ('top start') or at the bottom of the cycle ('bottom start').

Within each 20-trial block, each combination of starting position and cycling direction was performed for five consecutive trials (*Figure 1C*). Monkeys performed an average of 29 and 21 trials per condition per day (monkey E and F, respectively). Monkeys cycled quickly, with a median angular

speed of 2.2 cycles/s and 1.8 cycles/s (monkey E and F; *Figure 1D,E*). In contrast, the non-performing hand moved very little (leftmost distributions in *Figure 1D,E*). Mean angular speed for the non-performing arm was 0.0016 cycles/s and 0.024 cycles/s (monkey E and F).

## Neural and muscle responses

We analyzed the firing rates of 1150 units recorded from both hemispheres (*Figure 1F–G*). Most recordings were from M1 proper, and a few were from the adjacent caudal region of PMd. Trial-averaged firing rates were computed by filtering single-trial spike-trains to produce a continuous rate, and then adjusting the time-base to temporally align behavior across trials (*Figure 2*). Neural responses were typically rhythmic (*Figure 3E–H*), and could be nearly sinusoidal (*Figure 3E*) or could contain additional higher frequency structure (*Figure 3G*). For comparison, we recorded the activity of the major muscles in both arms (48 total recordings). The temporal features of individual-muscle responses (*Figure 3A–D*) in many ways resembled those of individual-neuron responses. However, muscles and neurons were quite different in the degree to which responses were restricted to movements of a single arm.

Muscles exhibited robust activity only when their driven arm performed the task (*Figure 3A–D*). For example the left anterior deltoid (*Figure 3A*) was active when the left arm performed the task (*blue*) but not when the right arm performed the task (*red*). While expected, this direct confirmation is important because of the possibility that muscles might have been active in ways that did not move the pedal (e.g. co-contraction). Such activity could potentially have been substantial, which would have complicated the interpretation of neural activity. A few muscles exhibited weak activity when the task was performed by their non-driven arm (*Figure 3A,C*). However, this typically occurred only at the end of movement, consistent with tensing to aid stability when stopping.

In contrast to the muscles, neurons were typically active throughout the movement, regardless of whether the task was performed with their driven (contralateral) or non-driven (ipsilateral) arm. A few neurons were active only when cycling with the driven arm (*Figure 3E*), and on rare occasions a neuron was active only when cycling with the non-driven arm (*Figure 3H*). However, most neurons were active in both situations (*Figure 3F,G*). Neural responses could be quite different when cycling with the driven versus non-driven arm. Neural response patterns could change in both phase (*Figure 3F*) and structure (*Figure 3G*) depending on which arm performed the task.

## Neurons are active during movements of either arm

To quantify the arm preference of individual units, we compared firing-rate modulation when the task was performed with the driven versus non-driven arm. Modulation was assessed as the standard deviation of the firing rate across times and conditions. Average modulation was computed once across all conditions where the driven arm performed the task, and again across all conditions where the non-driven arm performed the task. We analyzed only the middle cycles of movement (excluding the first cycle and the last two cycles). Focusing on this 'steady state' neural response aids interpretation because non-performing-arm muscles had minimal steady-state modulation. We computed an 'arm preference index': the difference in modulation for the driven versus non-driven arm, divided by the sum. This index ranges from −1 to 1, with the extremes indicating complete preference for the non-driven and driven arms respectively. An arm preference index of zero indicates that a neuron was equally responsive regardless of the arm used.

To establish a baseline for comparison, we computed the arm preference index for each muscle. Arm preference indices were typically high for the muscles (*Figure 4A*) confirming that muscles were active primarily when the task was performed with their driven arm. A few muscles showed weak activation regardless of the arm being used, resulting in lower indices. However, most muscles had robust responses during cycling, and were much more active when the task was performed with the driven arm. The median arm preference index was near unity (E: 0.86; F: 0.98; *Figure 4A*, *blue dots*) and the modal response occurred at unity.

In contrast, neurons rarely had arm preference indices near unity (*Figure 4B*). Instead, the distribution was centered only slightly above zero (median = 0.07 and 0.31 for Monkey E/F). Thus, neural responses were much more likely than muscle responses to be similar in magnitude regardless of the arm used. Furthermore, many neurons had arm preference indices < 0, indicating stronger modulation when the non-driven arm performed the task (Monkey E: 201/533 neurons; Monkey F: 107/617

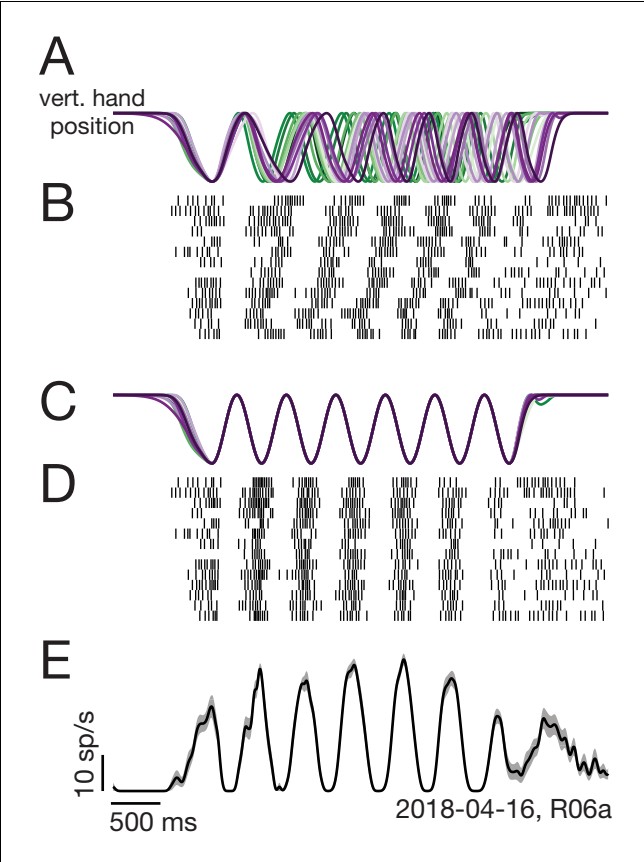

**Figure 2.** Two-step trial alignment procedure. (**A**) Vertical hand position on all trials (within one session) for an example condition. Data are aligned to the first half-cycle of movement. Trials are colored green to purple based on the average cycling speed for that trial. (**B**) Raster plot of spike times for an example neuron, for the same trials shown in A. Trials are ordered by average cycling speed and aligned as in A. (**C**) Hand position traces after the second alignment step: adjusting the time-base of each trial so that cycling during the middle six cycles matched the typical 2 hz pedaling speed. (**D**) Spike times after the second alignment step. (**E**) Average firing rate calculated after the second alignment step. Black: Mean firing rate. Gray shading: standard error across trials. Label at bottom right gives neuron identity.

DOI: https://doi.org/10.7554/eLife.46159.003

neurons). Results were virtually identical (median arm-preference indices of 0.06 and 0.31) if analysis was restricted to more posterior recordings (excluding the influence of any neurons recorded from PMd or from the border region between M1 and PMd).

Thus, neurons can be quite active even when the task is performed with their non-driven arm. Might such responses be related to small movements of the driven arm? This explanation is unlikely a priori. As described above, movements of the non-performing arm were small (*Figure 1D,E*) and corresponding muscle activity was weak (*Figure 4A*). In principle, neural responses related to weak muscle activity might be magnified via normalization or some other non-linearity. However, such magnification would need to be very strong. To match the median neural arm preference indices, muscle activity in the non-performing arm would need to be magnified by a factor of 12 (monkey E) and 52 (monkey F). Furthermore, magnification cannot account for the finding that neurons commonly had negative arm preference indices, while muscles rarely (monkey E) or never (monkey F) did.

We performed an additional control to ask whether the responses of neurons, when their non-driven arm performed the task, were in fact reflecting small movements of their driven arm. Such movements varied across trials (*Figure 5A*), allowing us to divide trials into those with movements larger than the median (very modest movement, *red*) versus lower than the median (nearly

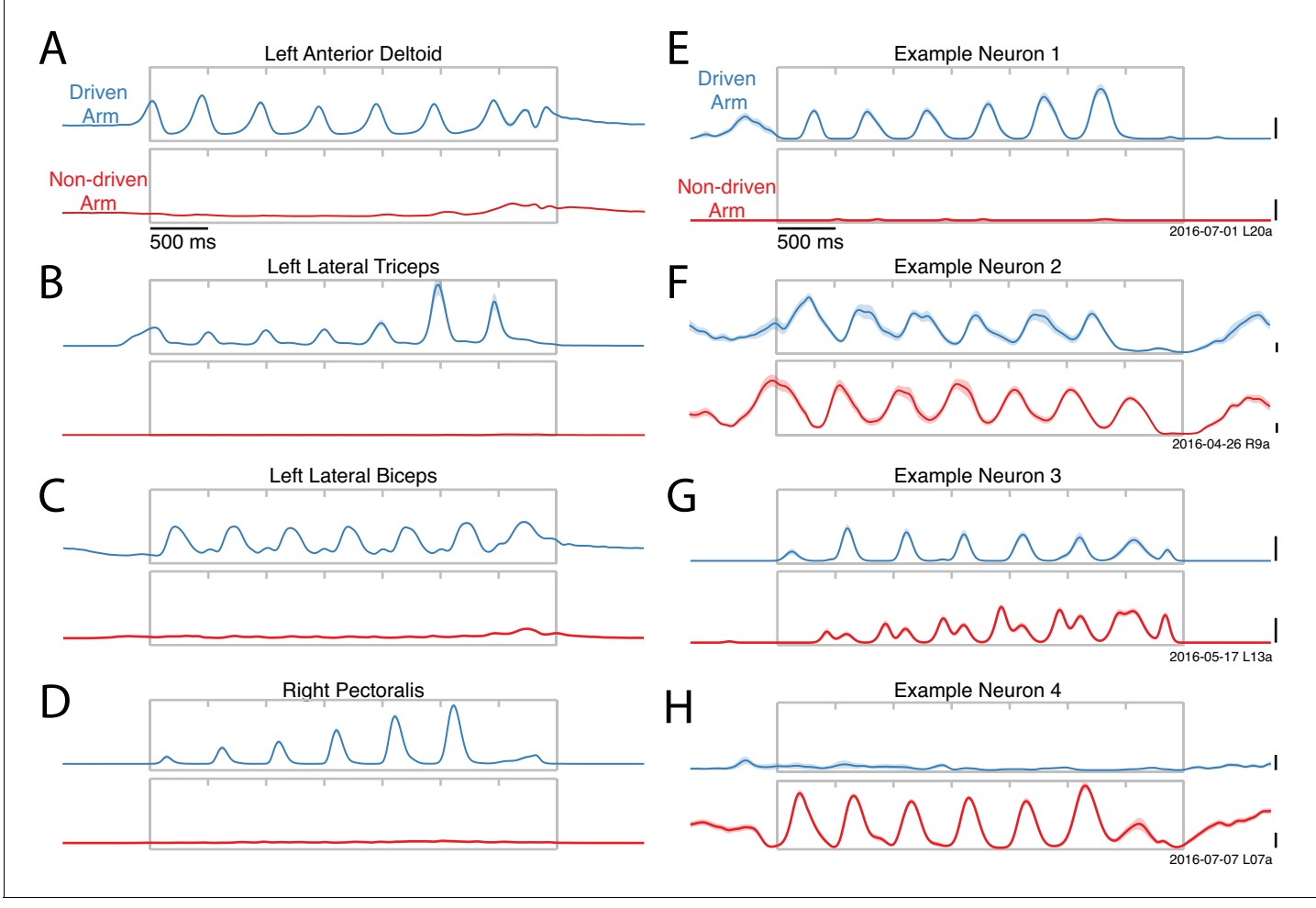

**Figure 3.** Activity versus time for four example muscles (**A–D**) and four example neurons (**E–H**). Each panel shows activity for one condition performed with either the driven arm (blue) or the non-driven arm (red). Each trace plots trial-averaged activity, with flanking envelopes (sometimes barely visible) showing standard errors. Gray boxes indicate when the pedal was moving, with tick marks dividing each cycle. All examples are from Monkey E. Example neural recordings were from single-unit isolations for all figures (population analyses include single- and multi-unit isolations). Calibration bars are 10 spikes/s. For the muscles, the vertical scale is arbitrary but conserved within each panel.

DOI: https://doi.org/10.7554/eLife.46159.004

stationary, *black*). Average firing rates were very similar in these two cases, as illustrated for one example neuron (*Figure 5B*). Differences were small and were significant at only a few moments (*black dots*) $p < 0.05$ across 1000 resamples, Materials and methods). Across all neurons, 5% of data-points showed significant differences (*Figure 5C,D*). This equals the percentage expected by chance, and is thus consistent with no reliable impact of small movements. Applying this same analysis to the (weak) muscle activity in the non-performing arm revealed significant differences at double the chance rate (10% of data-points) overall, and peaking at four times the chance rate near the end of the movement when muscle activity became more prominent (21% of data-points).

In summary, muscles were silent or at most weakly active when the task was performed with their non-driven arm. That weak muscle activity was statistically coupled to small movements of the driven (non-performing) arm. In contrast, most neurons exhibited robust responses when the task was performed with their non-driven arm. Those responses were not statistically linked to small movements of their driven (non-performing) arm. Prior studies have found that neurons can be active when a task is performed with their non-driven arm, although to varying degrees (*Cisek et al., 2003*; *Donchin et al., 1998*; *Kermadi et al., 1998*; *Steinberg et al., 2002*; *Tanji et al., 1988*). The present

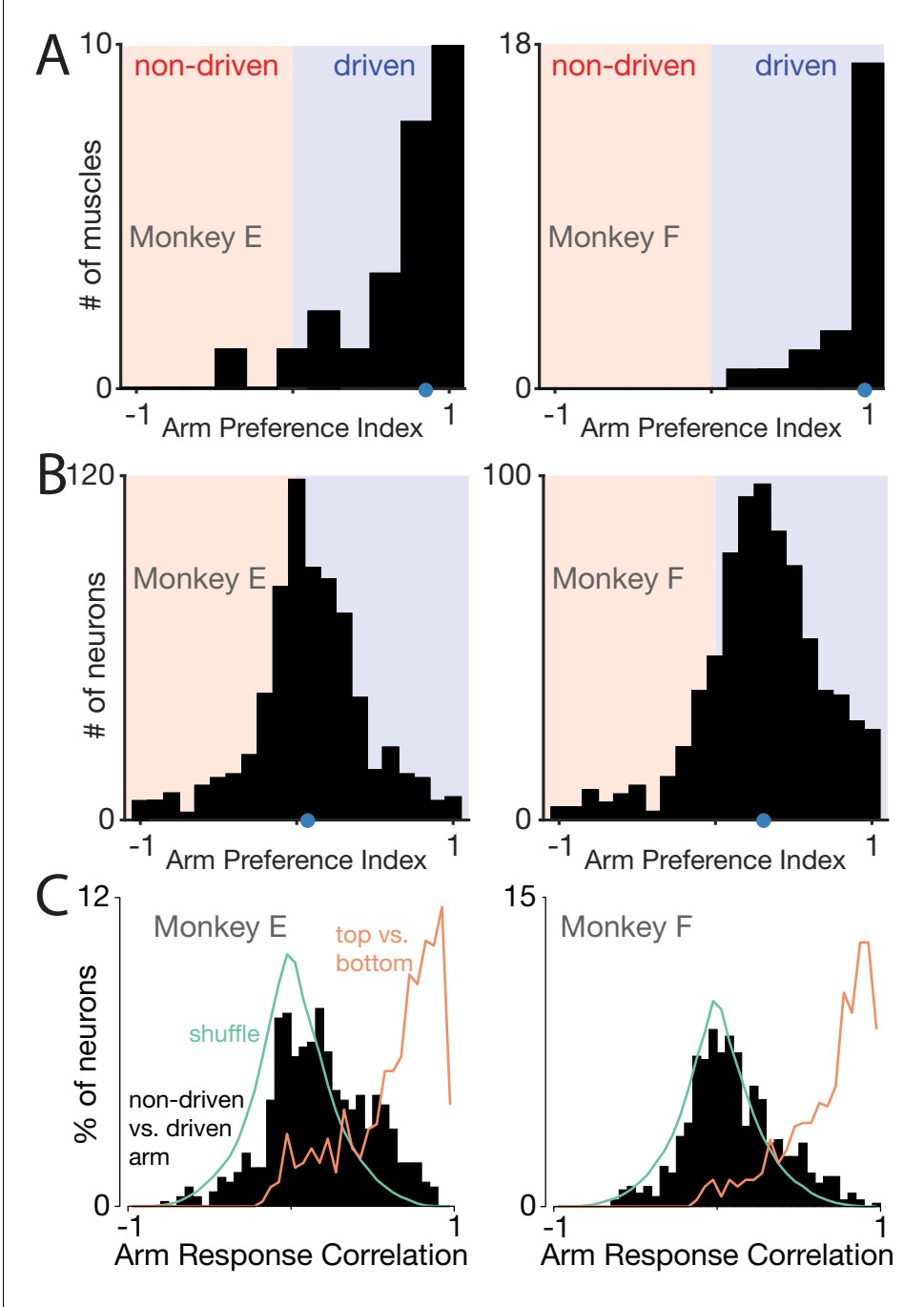

**Figure 4.** Muscle responses are lateralized while neural responses are not. (**A**) Histograms of arm preference index for all recorded muscles. Shaded regions indicate preference for the non-driven arm (red) and driven arm (blue). Blue dot indicates the median. (**B**) Histograms of arm preference index for all recorded neurons. (**C**) Histograms summarizing, for single neurons, similarity of responses when the task is performed with the driven versus non-driven arm. For each neuron, we computed the correlation between those two responses. Black histograms plot the distribution of such correlations across all neurons. Orange trace: control demonstrating that high correlations are observed, as expected, when comparing responses during top-start versus bottom-start conditions. Green trace: distribution of correlations between pairs of different units. This distribution illustrates the range of incidental correlations observed if there is no tendency for responses to be similar, other than being drawn from the same overall set of responses.

DOI: https://doi.org/10.7554/eLife.46159.005

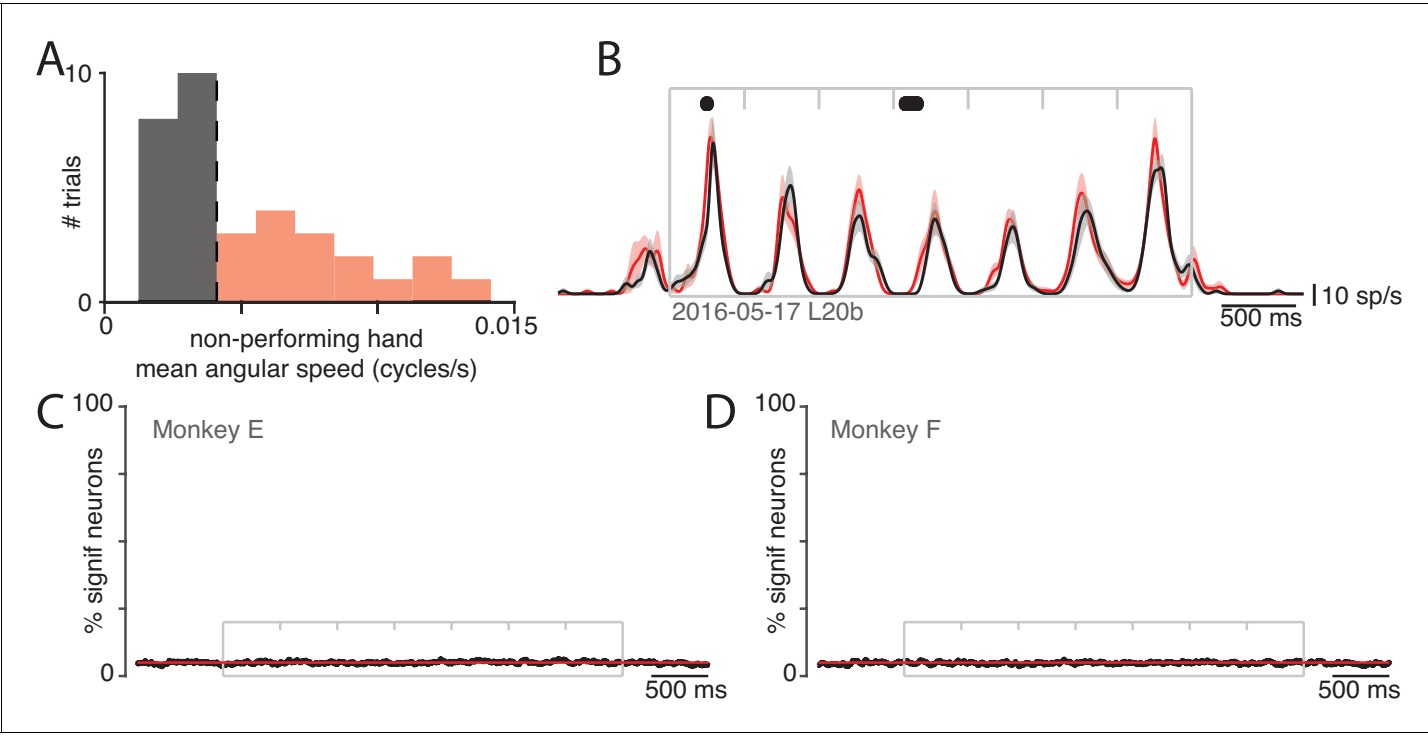

**Figure 5.** Small movements of the non-performing arm cannot explain modulation of neural activity within the non-driving cortex. (**A**) Analysis employed the distribution (across trials) of the mean angular speed of the non-performing arm. This distribution is shown for one condition, recorded on 1 day. Trials were divided into those with mean speed less than (gray) or greater than (red) the median (vertical dashed line). (**B**) Firing rate of one example neuron for these two groups: trials with speeds less than (black) and greater than (red) the median. Envelopes show standard errors of the mean. Black dots at top indicate times when the two rates were significantly different ( *p* < 0.05). Plotting conventions as in *Figure 3*. (**C**) Percentage of neurons (black trace) showing a statistically significant difference ( *p* < 0.05) in firing rate between trials with speeds less than versus greater than the median. Differences occurred as often as expected by chance (red line at 5%). Gray box denotes the time of movement. Each tick mark delineates a cycle. Data are for monkey E. Analysis is based on 426 units. (**D**) As in C but for Monkey F. Analysis is based on 479 units.
DOI: https://doi.org/10.7554/eLife.46159.006

findings replicate the finding of weakly lateralized responses in motor cortex, and largely rule out potential explanations based on small residual movements of the driven arm.

## Neural response patterns are limb-dependent

One plausible explanation for weakly lateralized responses is that neural activity encodes higher-level, limb-independent features of movement. For example, activity might encode hand velocity, movement goal, or some other quantity, regardless of which limb is moving. Preparatory activity in the more anterior rostral premotor cortex can exhibit largely limb-independent responses (*Cisek et al., 2003*). Might this also be true in motor cortex during movement? The two arms performed very similar movements in our task. Thus, limb-independence should be reflected by similar neural responses regardless of the performing arm. Instead, neural responses were strongly limb-dependent. Responses often differed in phase (*Figure 3F*) and/or structure (*Figure 3G*) depending on which arm performed the task.

To provide quantification, for each neuron we computed the correlation between the firing rate patterns when the driven versus non-driven arm performed the task (*Figure 4C*). Analysis considered only the middle cycles (2-5). This ensured that high correlations indicate similar response patterns, not simply firing rates that rise non-specifically during movement. On average the correlation was near zero (median correlation: 0.16 and 0.08 for Monkey E and F). Strongly correlated responses were the minority: only 18/533 (E) and 9/617 (F) neurons had correlations >= 0.75. Thus, for a given neuron, there was remarkably little relationship between responses when the task was performed with the driven versus non-driven arm.

For comparison, we computed the distribution of response correlations between random pairs of neurons. This yields a distribution of correlations expected by chance if responses are related only in the sense that they come from the same overall set of responses. The empirical distributions (*black*) were shifted significantly rightwards relative to these 'shuffled' (*green*) distributions ($p<10^{-3}$ for both monkeys, bootstrap, Materials and methods). Yet while significant, this shift was very modest: the empirical distributions were centered much closer to zero than to unity. Thus, the correlation between responses for the two arms, for a given neuron, was only slightly greater that the correlation between two arbitrarily chosen neurons.

Might correlations appear artificially low if responses are weak or noisy? While sampling error will inevitably reduce correlations, this is unlikely to be the source of the low correlations we observed. Cycling evoked particularly strong neural responses with correspondingly small standard errors of the mean firing rate (*Figure 3E–H*, envelopes show SEM). We further addressed this concern by computing, for each neuron, the correlation between the firing rate for top-start versus bottom-start conditions. Behavior was very similar for these two conditions during the middle cycles (after aligning phase), and correlations should thus be high. This was indeed the case (*Figure 4C*, *orange*, medians of 0.72 and 0.77), confirming that sampling error did not impede the ability to measure high correlations.

These results rule out the hypothesis of predominantly limb-independent responses. Individual-neurons showed very different responses depending on which arm performed the task. Responses were almost as different as if there were no relationship between the activity patterns associated with the two arms.

## Neuron-neuron correlations are limb-dependent

Consider two neurons that share a similar response pattern when the task is performed with the driven arm (*Figure 6A–D* plots four such example pairs). What occurs when the task is performed with the non-driven arm? From the analysis above, we know that the response pattern of each neuron is likely to change. Does this occur in a coordinated fashion, such that the two neurons remain correlated with one another? This would be consistent with the idea that neurons with related responses 'encode' related features, and continue to do so in new contexts. In fact, this property was rarely observed. We did occasionally observe neurons that were strongly correlated when the driven arm performed the task, and remained strongly correlated when the non-driven arm performed the task (*Figure 6A*). Yet, it was also common for correlations to invert (*Figure 6B*), for strong correlations to disappear (*Figure 6C*), or for neurons to undergo very different changes in response magnitude (*Figure 6D*).

We computed correlation matrices to assess such effects across the population. To aid visualization, we ordered neurons to group responses that were similar when the task was performed with the driven arm, resulting in a block structure (*Figure 6E–F*, *left*). We asked whether this correlation structure remained similar when the task was performed with the non-driven arm. Note that it is possible for the correlation matrix to remain identical, even if every neuron changes its response, so long as correlated neurons remain correlated. Instead, the correlation structure was dramatically altered. As a result, the original ordering no longer groups neurons with similar response properties (*right column* of *Figure 6E–F*).

This change in correlation structure was not due to correlations being largely spurious, as could occur if estimated firing rates were noisy. To investigate this possibility, we assessed the correlation structure between conditions where cycling started at the top versus bottom of the cycle. The correlation structure was very similar in these two cases (compare *middle* and *left columns* of *Figure 6E–F*). This finding also demonstrates that not every change in the task results in a change in the correlation structure; changing the starting position had relatively little impact, while changing the performing arm had a dramatic impact.

Each matrix in *Figure 6E–F* corresponds to a given condition (a starting position and cycling direction). We wished to summarize, across all such conditions, the degree to which correlations are or aren't preserved when the task is performed with one arm versus the other. To do so, for each condition and each pair of neurons we plotted their firing-rate correlation when the non-driven arm performed the task versus their correlation when the driven arm performed the task (to aid visualization, the scatterplots in *Figure 6G* show a random 1% of data-points; density plots in *Figure 6H* encompass all data-points). Preserved correlations would yield diagonal structure. In fact, there was

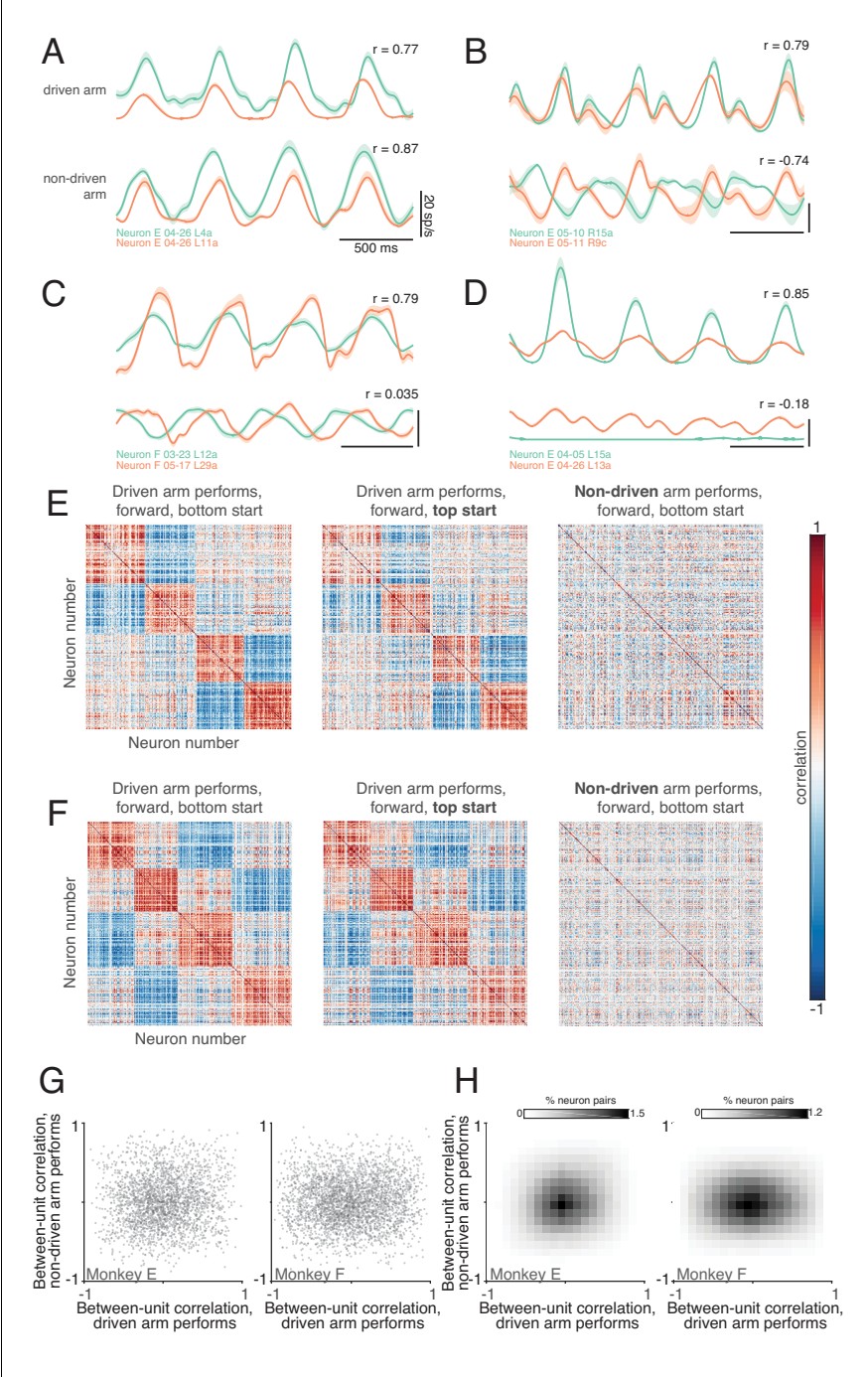

**Figure 6.** Correlations between neurons depend on which arm is used. (**A**) Average firing rates of two neurons (green and orange traces) during one condition, performed with the driven arm (top pair of traces) and non-driven arm (bottom pair of traces). Responses are shown for the middle cycles, which form the basis of the analysis below. For this example, the firing rates of the two neurons were strongly correlated when the driven arm performed the task, and remained so when the non-driven arm performed the task. Shading shows standard errors of the mean. (**B–D**) Responses of three other pairs of neurons. All pairs exhibit correlated responses when the task was performed with the driven arm. Correlations disappear or even invert when the task is performed with the non-driven arm. (**E**) Pairwise correlations between all units recorded from the left hemisphere of monkey E. Each panel plots the correlation matrix for one condition, indicated at top. Ordering of units was based on data in the left panel, and is preserved across panels. (**F**) same but for Monkey F. (**G**) Scatterplots of pairwise correlations. Each dot corresponds to a pair of units for a given condition, and plots the firing rate correlation when using the non-driven arm versus that when using the driven arm. To aid visualization, a randomly-selected 1% of data points are shown. (**H**) Density plot of pairwise correlations for Monkey E (left) and Monkey F (right).
DOI: https://doi.org/10.7554/eLife.46159.007

little tendency for correlated neurons to remain correlated, or for anti-correlated neurons to remain anti-correlated. The 'meta-correlation' was 0.1 and 0.05 (monkey E and F). Thus, if two neurons responded similarly when their driven arm performed the task, this said little regarding whether those neurons would respond similarly when their non-driven arm performed the task.

## Population activity is isomorphic across hemispheres

The above results demonstrate that individual-neuron responses and neuron-neuron correlations are very different depending on which arm performs the task. One potential explanation is that very different signals are present. Perhaps muscle-like signals are more prevalent when employing the driven arm while more abstract signals become prevalent when employing the non-driven arm. An alternative explanation is that many of the same signals are present, yet are reflected differently at the level of individual neurons.

We first asked whether muscle-like signals are present in both hemispheres. We trained a regularized linear decoder to predict performing-arm muscle activity based on neural activity (*Figure 7B*). We compared predictions based on activity in the driving cortex (contralateral to the performing arm) versus non-driving cortex (ipsilateral to the performing arm). Decoder performance was quantified by testing generalization for a held-out condition, repeating this procedure for each condition.

Muscle activity was accurately predicted by both driving-cortex activity (*Figure 7A*, *blue*) and non-driving-cortex activity (*red*). Across all conditions, generalization $R^2$ was high for both the driving and non-driving cortex (*Figure 7C,D*, generalization performance computed across all muscles). Generalization performance was lower for the non-driving cortex, but this was a small effect and was significant for only one monkey ($p = 0.15$ and $p = 0.016$ for monkey E and F, two-sided Wilcoxon signed rank test across eight conditions). We thus saw no evidence that signals related to muscle activity were absent in the non-driving cortex.

Might other signals be restricted to the driving cortex? Rather than attempting to infer specific candidate signals, we developed a method to address this question generically. We assessed how well driving cortex activity could be predicted by the dominant signals from the non-driving cortex. To do so, we projected non-driving cortex activity onto its top principal components, and then regressed driving cortex activity against these projections (*Figure 7E*). For comparison, we computed how well driving cortex activity could predict itself, by regressing the activity of each driving-cortex unit against principal component projections derived from all other driving-cortex units, in a leave-one-out fashion.

Generalization performance depended on the number of principal components (*Figure 7F,G*) and was higher for more components (indicating that overfitting was not a large concern). The key question is whether, for a given number of components, it matters whether driving-cortex activity is explained by driving-cortex projections versus non-driving-cortex projections. This was indeed the case, but it was a very modest effect. For example, when using 10 principal components, generalization $R^2$ was 82% and 85% (monkey E and F) when regressing against driving-cortex projections, versus 81% and 81% when regressing against non-driving-cortex projections. This difference was statistically reliable ($p < 0.01$ for both monkeys, randomization test with 100 resamples, Materials and methods) but much smaller than expected if there were signals that were strongly present in the driving cortex but weak or absent in the non-driving cortex (*Figure 7 Supplement 1*).

Thus, the dominant signals – those present in the top handful of principal components – are nearly identical between the driving and non-driving cortices. A caveat is that smaller signals could be quite different, which would be difficult to discern given our relatively simple task. Activity spans a modest number of dimensions when cycling at steady-state (five principal components account for ~80% of the variance). It is quite possible that differences in smaller signals, carried by other dimensions, could be uncovered with added task complexity (e.g. cycling at different speeds or under different loads).

## Neural trajectory tangling is low for both hemispheres

We recently described a major difference between M1 population activity and downstream muscle activity (*Russo et al., 2018*). Neural population activity, but not muscle population activity, avoids 'trajectory tangling.' Trajectory tangling is defined as the occurrence of similar population states with very different derivatives. Trajectory tangling becomes high if the population trajectory crosses

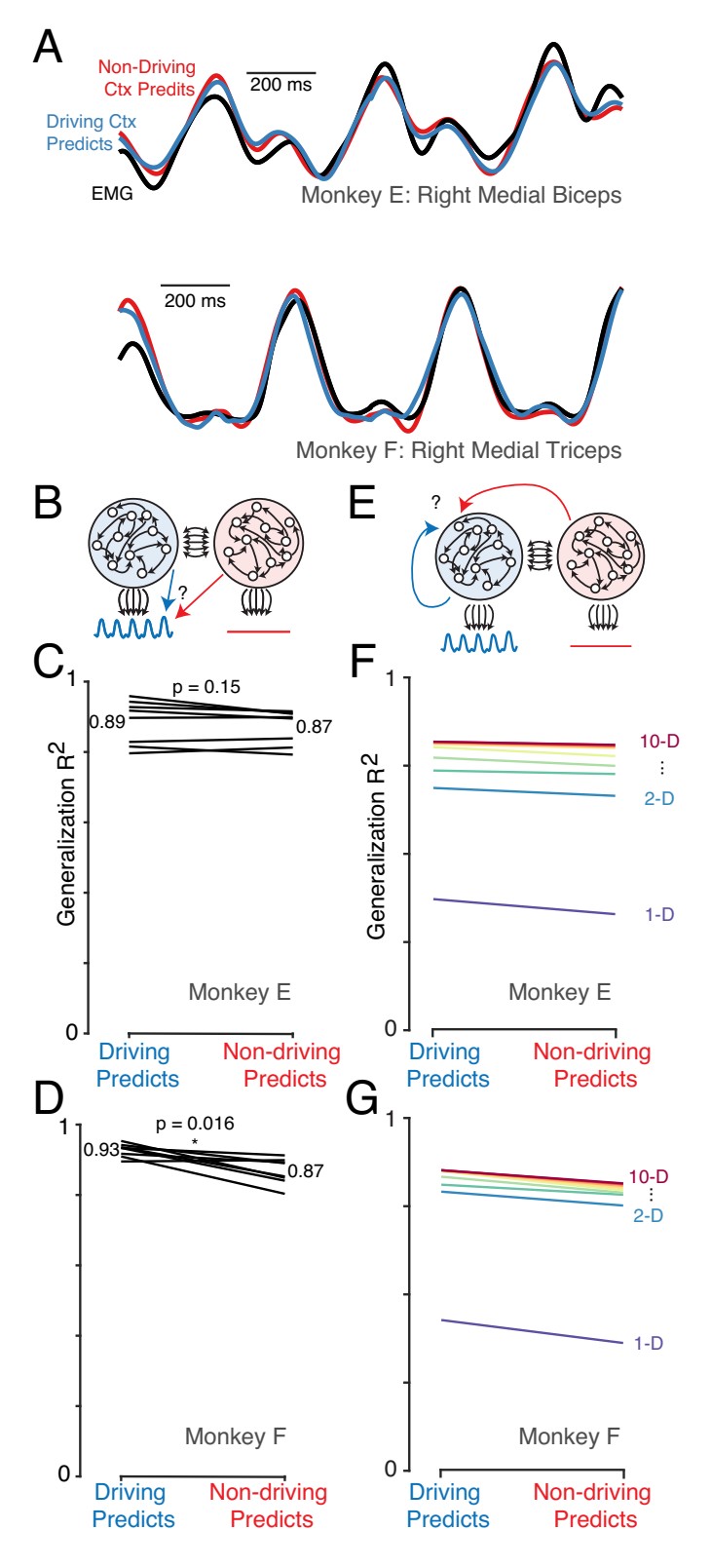

**Figure 7.** The dominant signals are very similar in the driving and non-driving cortex. (**A**) Example muscle activity patterns (black) and predictions of muscle activity based on a linear decode of neural activity in the driving (blue) and non-driving (red) cortex. Examples are shown for one muscle from each monkey. In both cases, data is from a test condition and illustrates generalization performance. (**B**) Cartoon illustrating the analysis approach for comparing muscle-decode performance from the driving versus non-driving cortex. Performing-arm muscle activity (blue trace) is a product of

*Figure 7 continued on next page*

*Figure 7 continued*

descending connections (black arrows) from neurons within the driving cortex (blue network). It should thus be possible to predict (blue arrow) that muscle activity from neural activity recorded from the driving cortex. The presence or absence of muscle-like signals within the non-driving cortex (red network) was assessed by asking how well such activity predicted (red arrow) performing-arm muscle activity. (C) Prediction performance for the comparison illustrated in B. Data are for Monkey E. Each line corresponds to one behavioral condition and shows the percent variance (of muscle population activity) predicted by decodes based on driving and non-driving cortex neural activity. (D) As in C, for Monkey F. (E) Cartoon illustration of analysis asking whether the signals carried by the driving cortex are also present in the non-driving cortex. When the task is performed with a given arm, neural activity within the corresponding driving cortex is predicted either from the activity of other neurons either within the driving cortex (blue arrow) or within the non-driving cortex (red arrow). (F) Prediction performance for the comparison illustrated in E. Data are for Monkey E. Each color trace shows the average generalization performance across eight behavioral conditions for a given projection rank, using rainbow colors from red (10 dimensions) to purple (one dimension). Each line compares performance when predicting driving cortex activity from driving cortex activity (left) versus from non-driving (right). (G) As in panel F, for data from Monkey F.

DOI: https://doi.org/10.7554/eLife.46159.008

The following figure supplement is available for figure 7:

**Figure supplement 1.** Confirmation that the analysis in *Figure 7C,D* is sensitive to signal mismatch between driving and non-driving cortices, if that situation were present.

DOI: https://doi.org/10.7554/eLife.46159.009

itself, or if the trajectory for one condition traverses near that for another condition while traveling in a different direction. Pattern-generating recurrent networks are noise-robust only if trajectory tangling is low, suggesting an explanation for why low trajectory tangling was observed in M1. It is unknown whether the non-driving cortex participates (via callosal connections) in pattern generation. Notably, it is possible for a cortical area to be active during cycling yet have high trajectory tangling; this was true of proprioceptive primary somatosensory cortex (*Russo et al., 2018*).

We computed the tangling index (*Russo et al., 2018*) for every timepoint during the middle cycles, across all conditions. The muscles often showed high trajectory tangling, revealed by a long right tail in the cumulative distributions (*Figure 8A,B*, *black lines*). The driving cortex displayed

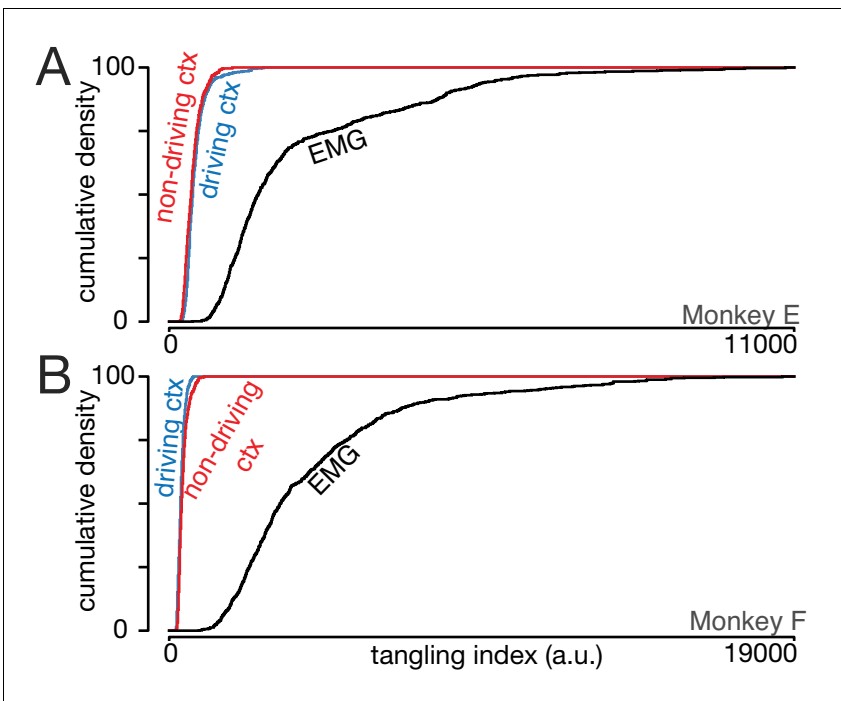

**Figure 8.** Trajectory tangling is similar for the driving and non-driving cortices. (A) Cumulative distribution of trajectory tangling for the driving cortex (blue), non-driving cortex (red), and muscle activity (black). Distributions were compiled across time points and conditions. Data for Monkey E. (B) As in A, for Monkey F.

DOI: https://doi.org/10.7554/eLife.46159.010

consistently low trajectory tangling: cumulative distributions (*blue*) plateaued early. This replicates prior results: trajectory tangling is much lower for the driving cortex than for the downstream muscle population. Notably, this is true even though individual-neuron and individual-muscle responses are superficially similar.

Trajectory tangling was also low for the non-driving cortex (*red*). For monkey E, tangling was slightly higher in the non-driving versus driving cortex (468 ± 201 versus 420 ± 153; mean ±S.D.) while the opposite was true for monkey F (374 ± 105 versus 430 ± 146). Thus, trajectory tangling is similarly low for both cortices, with only small and inconsistent differences. Critically, for both the driving and non-driving cortex, neural trajectory tangling was much lower than muscle trajectory tangling. The latter averaged 2296 ± 1766 for Monkey E and 4392 ± 2950 for Monkey F.

The presence of similarly low tangling in the driving versus non-driving cortex agrees with the finding above that the major signals are similar. Thus, there is little hemispheric difference regarding either the major signals or the organization of population trajectories. Notably, such similarity is not inevitable when comparing population responses; muscle trajectories had a different, and much more tangled, organization.

## Neural activity occupies different dimensions for movements of different arms

The above results document that during movement of a given arm, neurons in both hemispheres are active and carry similar signals. Indeed, one would rarely be able to tell, from the response of a given neuron, the hemisphere in which that neuron resides. On the one hand, this broad sharing of information makes sense given the heavily interconnected nature of the two hemispheres. On the other hand, the descending control provided by each hemisphere must still be able to separate signals related to the relevant arm, and avoid contamination from signals related to the other arm. Is the population response structured to facilitate that separation? Can separation be achieved based solely on the properties of the responses themselves, without knowing the hemispheric identity of each neuron?

In confronting these questions, we took inspiration from recent work suggesting that some neural dimensions in motor cortex are 'muscle potent' – activity in those dimensions produces output that will influence the muscles – while other dimensions are 'muscle null' – activity in those dimensions has no direct outgoing impact on muscle activity (*Druckmann and Chklovskii, 2012*; *Kaufman et al., 2014*). The presence of output-null dimensions is natural (and typically inevitable) in pattern-generating recurrent networks. We wondered whether this principle might apply to the present case. We considered all recorded neurons, across both hemispheres, as a unified population. We asked whether signals related to the movement of each arm are partitioned in a manner that could allow signals related to one arm to naturally avoid impacting the other arm.

We applied principal component analysis (PCA) to the population response (the trial-averaged firing rate of every unit) during the middle cycles of each behavioral condition. PCA was applied separately for conditions where the task was performed with the left versus right arm. For example for forward cycling starting at the top, PCA was applied once when cycling with the left arm (yielding a 'left-arm' subspace) and again when cycling with the right arm (yielding a 'right-arm' subspace). Importantly, in both cases, PCA considered the responses of the same unified population of neurons (all recorded neurons across both hemispheres). The key question was to what degree the left-arm subspace, constructed without any reference to activity when the task was performed with the right arm, did or did not capture such activity (and vice versa for the right-arm subspace).

Activity from a held-out condition was used to test how well left- and right-arm subspaces captured responses when each arm performed the task. For example, *Figure 9A* employs a left-arm subspace, based upon responses when the left arm performed a particular condition (forward cycling, starting at the top). This same space captured a robust response when the left arm performed the held-out condition (*blue* trajectory; forward cycling, starting at the bottom) but not when the right arm performed that same held-out condition (*red* trajectory). We quantified this effect by computing the cumulative variance explained by the left-arm space, as a function of the number of principal components used. Cumulative variance explained rose rapidly (as expected) when considering held-out conditions performed by the left arm (*Figure 9B*, *blue* traces), but not when considering held-out conditions performed by the right arm (*red* traces).

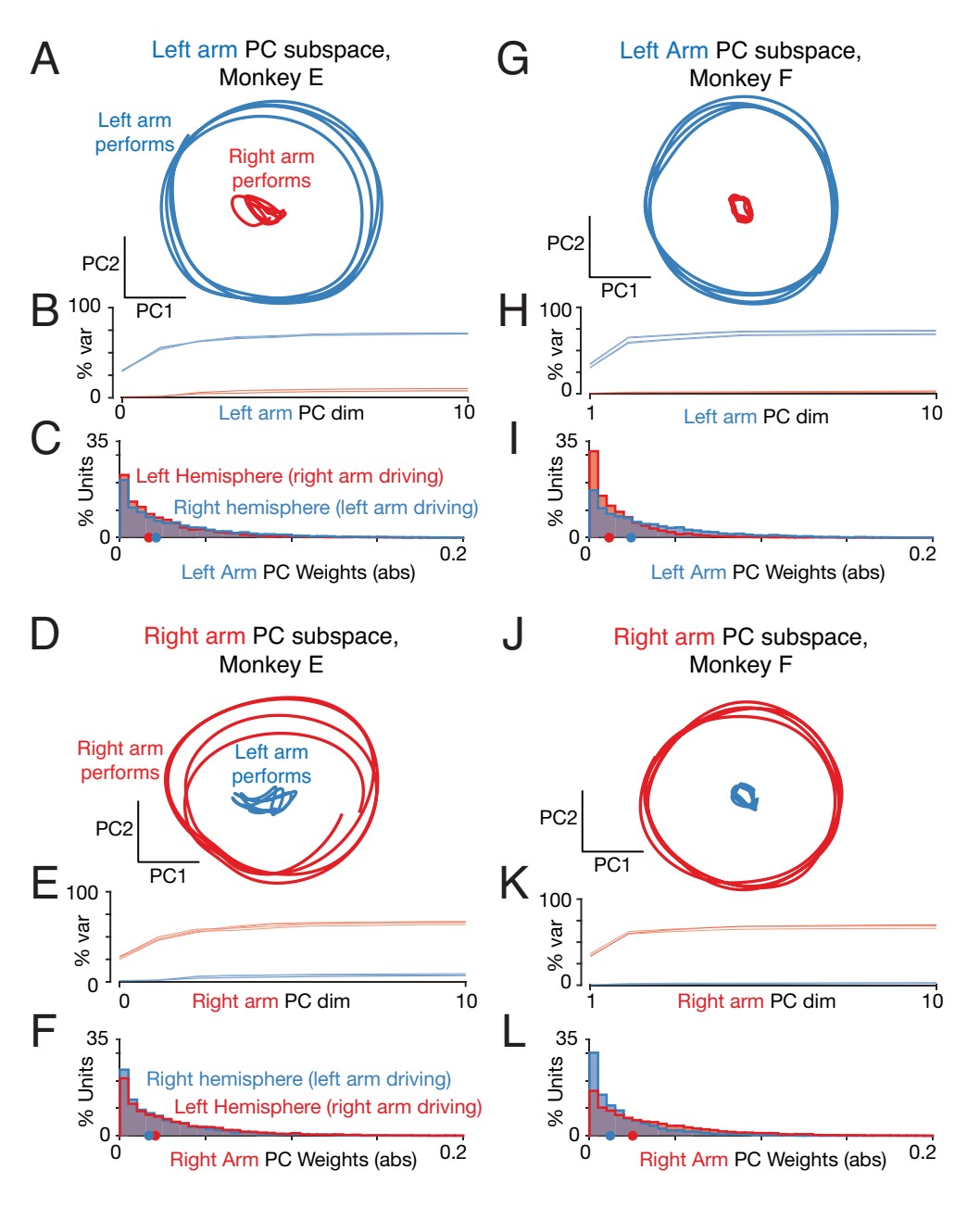

**Figure 9.** Left-arm and right-arm related activity lie in orthogonal subspaces. (**A**) Projection of population activity onto the first two dimensions of the left-arm subspace. The left-arm subspace was found based on data from one condition (forward, top-start) performed with the left arm. This same subspace captured responses during another condition (forward, bottom-start) performed by the left arm (blue trajectory), but not for a condition (forward, top-start) performed with the right arm (red trajectory). Analysis considers data from the middle cycles, and all recorded neurons from both hemispheres. Data are for monkey E. (**B**) Cumulative percent variance explained by the left-arm subspace, for responses recorded when the left (blue) or right (red) arms performed the task. Variance explained was always computed for the left-out conditions (conditions not used to find the space). The analysis was performed with each condition serving as the training condition (used to find the space) once. There were four such conditions (two directions by two starting positions), yielding four (largely overlapping) traces of each color. Data are for monkey E. (**C**) Magnitude of the contribution to the left-arm subspace from units in the right (driving) and left (non-driving) hemispheres. A rightwards shift would indicate greater contribution from that hemisphere. Distribution medians are indicated by red and blue dots. The distribution was based on the absolute value of the weights contained in the top five PCs (after which the percent variance captured had largely

*Figure 9 continued on next page*

*Figure 9 continued*

plateaued). Histograms are combined across all behavioral conditions, so each neuron contributes 4 conditions x 5 PCs = 20 weights. (**D–F**) As in A-C, for the 'right arm' subspace. Note that the data projected are the same as in panel A – only the subspace differs. (**G–L**) As in A-F, for Monkey F.

DOI: https://doi.org/10.7554/eLife.46159.011

The following figure supplements are available for figure 9:

**Figure supplement 1.** Repetition of analysis in *Figure 9*, but analyzing the neural population from the right cortex only.

DOI: https://doi.org/10.7554/eLife.46159.012

**Figure supplement 2.** Repetition of analysis in *Figure 9*, but analyzing the neural population from the left cortex only.

DOI: https://doi.org/10.7554/eLife.46159.013

---

Analogous results were observed for the right arm (*Figure 9D,E*) and for the other monkey (*Figure 9G,H,J,K*). It was always the case that relatively little variance was captured when activity for right-performing conditions was projected onto the left-arm space, and vice versa. Averaged across all such conditions, the top five principal components explained only 7 ± 1% (monkey E) and 2 ± 0.5% (monkey F) of the response variance (mean ±std. computed across eight conditions). This was in contrast to the considerable variance captured for held-out conditions performed by the same arm on which the space was trained: 65 ± 3% (monkey E) and 69 ± 2% (monkey F).

Thus, neural responses related to the two arms occupy nearly orthogonal subspaces. Dimensions that robustly capture activity when one arm performs the task do not continue to do so when the other arm performs the task. This was true both when analyzing a unified population (*Figure 9*) and when analyzing each hemisphere independently (*Figure 9—figure supplement 1,2*).

Orthogonal subspaces could occur trivially if neurons fall into two groups: neurons that respond only when the right arm performs the task, and neurons that respond only when the left arm performs the task. However, as documented above (*Figure 4B*) most neurons responded when the task was performed with either the left or right arm, with only a modest tendency for responses to be stronger when using the contralateral arm. Consistent with this, both subspaces involved contributions from neurons in both hemispheres (*Figure 9C,I,F,L*) with a modestly larger contribution from the driving cortex (e.g. the right hemisphere for the left-arm subspace) as expected given the modestly larger individual-neuron responses in the driving cortex.

## Neural trajectories on single trials

Is it also true that single-trial activity avoids the subspace associated with the non-performing arm? Presumably that subspace must be occupied at least weakly on single trials; the non-performing arm does occasionally move (if only very slightly). The key question is whether the occupation is more than modest, which is possible if trial-averaged data are unrepresentative of what occurs on single trials. To ask whether trial-averaged firing rates are representative, we first considered across-trial variability at the individual-neuron level. Studies have typically found that the Fano factor (the ratio of spike-count variance to spike-count mean) in cortex is approximately unity when an animal is fully engaged in a task and performing consistently (*Churchland et al., 2010b*). The presence of additional trial-to-trial 'network level' variability inflates the Fano factor above unity (*Churchland and Abbott, 2012*; *Litwin-Kumar and Doiron, 2012*; *Shadlen and Newsome, 1998*). During cycling, the Fano factor was less than or equal to one (0.99 and 0.69 for Monkey E and F, averaged across conditions), arguing against considerable network-level variability.

We next examined single-trial variability at the population level. We used a recently developed method: Latent Factor Analysis via Dynamical Systems (LFADS) (*Pandarinath et al., 2018*) LFADS employs latent non-linear dynamics to produce single-trial firing rates that explain the observed spiking. LFADs is not guaranteed to identify all forms of trial-to-trial variability but has been demonstrated to successfully infer meaningful variations in single-trial activity under a variety of circumstances. We used LFADS to infer single-trial firing rates, employing sessions where a reasonable number of neurons were recorded simultaneously (20-41). Estimated single-trial firing rates were then projected into the left-arm and right-arm subspaces (computed from trial-averaged rates). Single-trial trajectories had modest variability, but it was nevertheless the case that the left-arm

space (*Figure 10A,B,E,F*) captured considerable variance for left-arm performing conditions (*blue*) but not right-arm performing conditions (*red*). Analogous results were found for the right-arm subspace (*Figure 10C,D,G,H*). Thus, separation of left-arm and right-arm-related signals into different subspaces was present at the single-trial level. Separation was perhaps slightly weaker (e.g. in *Figure 10C*, the blue trace has non-negligible variance) than when analyzing trial-averaged rates, but this is expected when analyzing fewer neurons (*Figure 10—figure supplement 1*).

## Linear decoders naturally separate signals related to the two arms

The separation of activity into orthogonal subspaces could allow descending control of one arm to naturally ignore signals related to the other arm. To test the plausibility of this hypothesis, we trained linear decoders to predict muscle activity for a given arm, based on the activity of the entire neural population across both hemispheres. The decoder was trained using only conditions where that arm performed the task. For example, the decoder was trained to predict muscle activity in the right arm while the right arm performed the task. Restricted to this situation, decoders performed well, predicting a median of 91% (Monkey E) and 93% (Monkey F) of the variance on held-out conditions. Examples of predicted muscle activity (*Figure 11A,G*, *orange traces at top*) are shown for one muscle for each monkey. Decoders leveraged activity in both hemispheres (*Figure 11E,K*), as expected given that neurons in both hemispheres are active and predictive of muscle activity (as documented above).

The key question regards what happens when decoders are asked to generalize to conditions where the other arm performed the task. For example, does a decoder trained to predict activity in the right triceps, when the right arm performs the task, successfully predict little modulation of the triceps when the left arm performs the task? This is a stringent test of generalization both because it involves conditions very different from the training conditions, and because the neurons upon which the decode is based are active regardless of the arm used.

Decoders accurately generalized, and correctly predicted little modulation of muscle activity when the arm was not used to perform the task (*Figure 11A,G*, *orange traces in bottom subpanels*). To provide quantification, we computed the degree of modulation (the standard deviation of activity over time) of the actual and decoded muscle activity. Actual muscle modulation declined (as expected) when the relevant arm changed from performing to non-performing (*Figure 11B,H*, *black*). Decoded muscle modulation underwent a similar decline (*orange*). The decline in decoded muscle activity was significant for both monkeys (Monkey E: $p = 1.8 \times 10^{-17}$, $n = 96$; Monkey F: $p = 1.8 \times 10^{-17}$, $n = 96$; Wilcoxon signed-rank test across muscles and conditions).

We further documented this effect by computing distributions (*Figure 11C,I*) of actual (*gray*) and predicted (*orange*) muscle modulation. Modulation was strong when the relevant arm (the arm that contained the muscle in question) was the performing arm (*top sub-panels*). Modulation was much weaker when that arm was the non-performing arm (*bottom sub-panels*). Notably, this occurred for decoded muscle activity, even though the decoder was not trained on those non-performing conditions.

The ability of the decoder to ignore activity related to the 'wrong' arm is inherited from the orthogonality of right-arm and left-arm subspaces. When trained using right-arm conditions, decoders naturally employ dimensions within the right-arm subspace. Because those dimensions are largely unoccupied when the left arm performs the task, the decode shows minimal modulation. Due to these properties, decoders naturally produce predicted muscle activity with positive arm-preference indices (*Figure 11D,J*, *orange histograms*). These distributions are right-shifted relative to those for the neural activity upon which decoding was based (*red histogram*). Thus, the structure of population activity ensures that a decoder, trained to extract activity related to one arm, will naturally tend to ignore activity related to the other arm.

Yet while decoders tended to naturally ignore activity related to the 'wrong' arm, this feature was imperfect: small amounts of residual modulation were still present *Figure 11A,G bottom subpanel*). Overall, predicted EMG modulation in the non-performing arm (*Figure 11C,I*, *orange* distribution in bottom subpanel) was significantly higher than the true EMG modulation (Monkey E:$p = 0.0013$; Monkey F: $p < 0.001$; Wilcoxon signed-rank test across 96 muscles and conditions). This led to arm-preference indices, for the decoder, that were smaller than those of the muscles (*Figure 11D,J*,

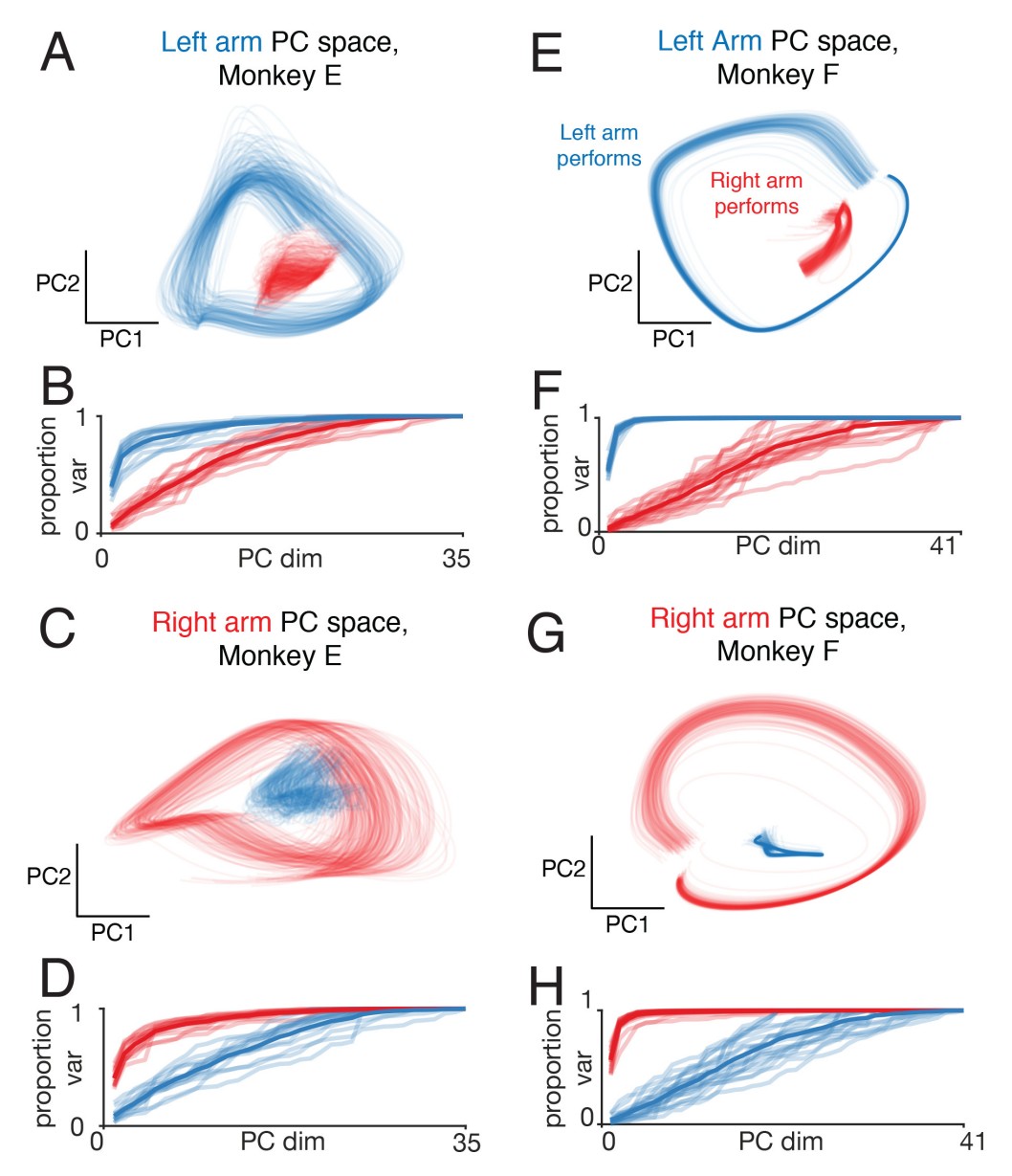

**Figure 10.** Left-arm and right-arm related activity on single trials. (**A**) Similar analysis to that in *Figure 9A*, but projecting single-trial firing rates inferred using LFADS. Each trace plots the projection of the population response during one cycle of one trial (both top-start and bottom-start conditions are shown). Data are for one session performed by monkey E. (**B**) Cumulative proportion of variance explained by the left-arm subspace, for single-trial responses recorded when the left (blue) or right (red) arms performed the task. Each recording session contributes four traces: one for each pedaling direction and performing arm. The dark trace shows the average. The proportion of variance was normalized to always be unity for 35 PCs (corresponding to the maximum population size across all included datasets for this monkey), to allow comparison across sessions with different numbers of isolations. (**C**) The same example trials shown in A, projected onto the first two dimensions of the right-arm subspace. (**D**) As in B, for cumulative percentage variance explained by the right-arm subspace. (**E–H**) As in A-D, for Monkey F.

DOI: https://doi.org/10.7554/eLife.46159.014

The following figure supplement is available for figure 10:

**Figure supplement 1.** High cell-counts help to reliably reveal when neural subspaces are orthogonal.
DOI: https://doi.org/10.7554/eLife.46159.015

*orange* distributions lie to the left of *black* distributions). Is this a fundamental limitation of a linear decoder or does it simply reflect the challenge of generalizing to a very different situation?

To address the second possibility, we trained decoders using both left-arm-performing and right-arm-performing conditions (with generalization tested using left-out conditions of each type). This train-both strategy led to further improvement in the ability to ignore neural activity related to the 'wrong' arm (*Figure 11*, *green traces and distributions*). For example, the green distributions in *Figure 11C,I* are shifted leftward of the orange distributions. ($p = 0.0013$ and $p<0.001$ for the two monkeys, Wilcoxon signed-rank test across 96 muscles and conditions). Yet, decoded muscle activity was still modulated slightly more than observed empirically (Monkey E: $p = 0.01$, Monkey F: $p<0.001$). As a result, train-both decoders predicted muscle activity with arm-preference indices much higher than that of the neural activity, but not quite as high as those of the muscles themselves (*Figure 11D,J*). This remaining modulation of the decode likely reflects the use of a linear approximation to the non-linear function relating neural and muscle activity, and/or the incomplete sampling of the neural population.

The above results demonstrate that even simple linear decoders can largely ignore activity related to the wrong arm. This ability can be improved by directly training decoders to do so. Yet, even without such training, decoders successfully generalize and predict little muscle activity when the relevant arm is not performing the task. There is thus no paradox in the presence of robust neural activity across both hemispheres, despite the absence of muscle activity in the non-performing arm.

## Discussion

Neural signals related to movements of the right and left arms were mixed across hemispheres; when one arm moved, neurons in both hemispheres were modulated. Signals were also mixed across neurons; most neurons responded when both their driven and non-driven arm performed the task. Individual neurons responded very differently depending on which arm was moving. Yet at the level of the population, both hemispheres contained similar information. Surprisingly, we did not find signals that were strongly present in the driving cortex but absent in the non-driving cortex. This was true even for muscle-like signals, which could be decoded similarly well from either hemisphere.

Despite this intermixing, signals corresponding to the two arms were naturally separable at the level of the neural population. Activity related to the two arms occupied nearly orthogonal subspaces. As a result, even simple linear decoders could read out commands for one arm while ignoring commands for the other arm. Indeed, decoders naturally ignored activity related to the 'wrong' arm, even without being trained to do so.

### Separation of information across dimensions is a common feature of cortical activity

Our results contribute to an increasingly broad set of studies reporting that neural activity related to different computations or task parameters is often separated across neural dimensions, instead of at the level of brain areas or individual neurons (*Druckmann and Chklovskii, 2012*; *Machens et al., 2010*; *Mante et al., 2013*; *Raposo et al., 2014*). For example, during reaching, dimensions carrying preparatory activity are orthogonal to dimensions carrying movement-related activity generally (*Elsayed et al., 2016*) and muscle-related signals specifically (*Kaufman et al., 2014*). Preparatory dimensions are thus proposed to be 'muscle null' – with no net drive to downstream muscles. Muscle-null dimensions are also expected during movement; they are essential in network-style models of reach generation (*Churchland et al., 2012*; *Sussillo et al., 2015*) and to the hypothesis that population activity is structured to avoid trajectory tangling (*Russo et al., 2018*). Our results are consistent with the hypothesis that right-arm-related neural activity falls in dimensions that are muscle-null with respect to the left arm, and vice versa. This hypothesis makes a prediction that could be tested with simultaneous EMG and large-scale neural recordings: small contractions within the non-performing arm should be correlated with small excursions into the corresponding subspace.

Although we do not yet know the mechanisms by which the orthogonality of different signals is established, it is noteworthy that two randomly oriented dimensions are essentially guaranteed to be nearly orthogonal in a high-dimensional space. Thus, if a learning mechanism ensures that

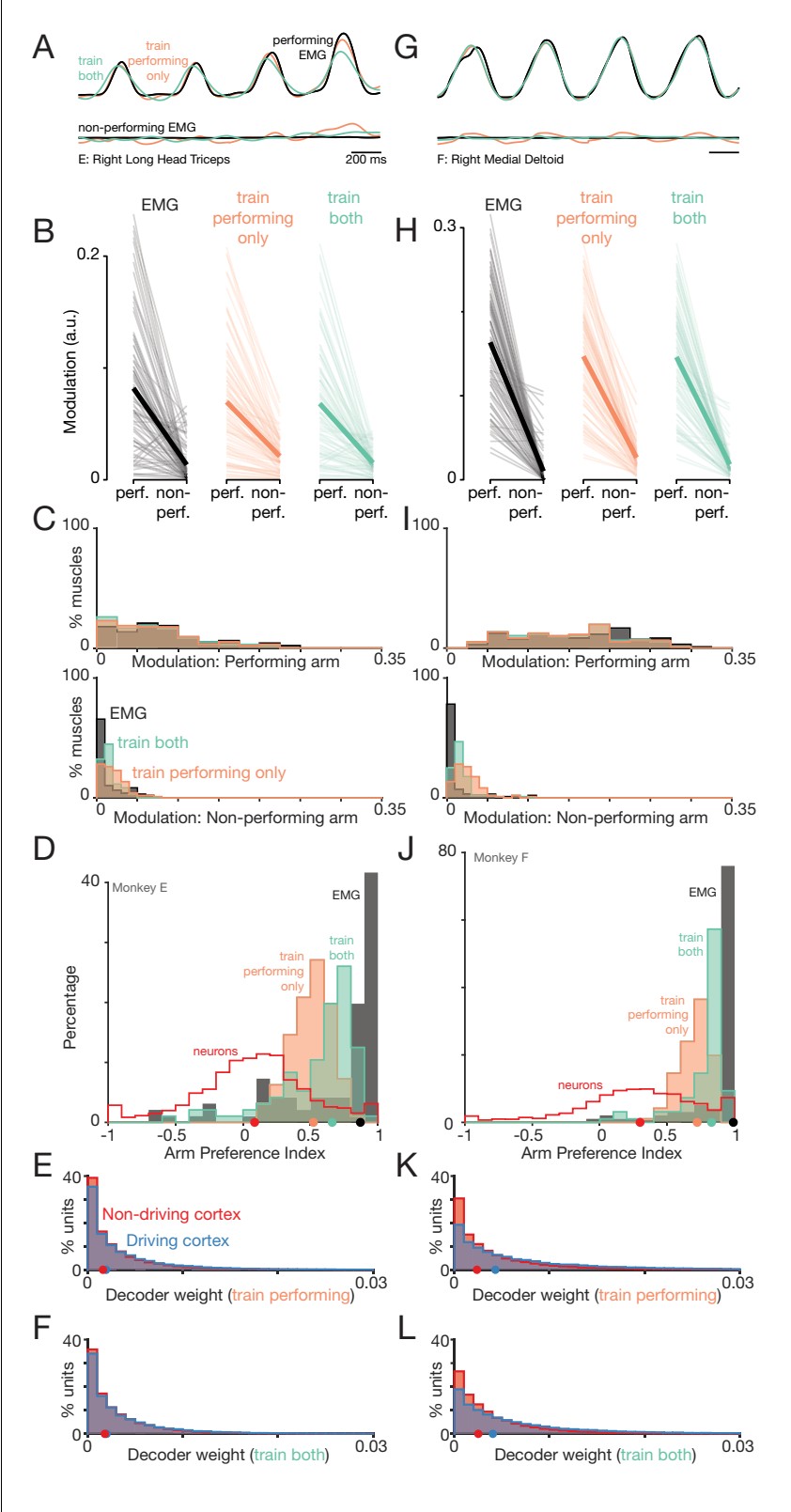

**Figure 11.** Muscle-activity decoders ignore activity related to the 'wrong' arm. (**A**) Actual (black) and predicted activity of the long head of the triceps, recorded from the right arm of monkey E. Top traces: activity while the right arm performed the task. Bottom traces: activity while the left arm performed the task. Orange traces: Predictions of a decoder trained only on right-arm-performing conditions ('train-performing-only' decoder). Green traces: Predictions of a decoder trained on a subset of both left- and right-arm-performing conditions ('train-both' decoder). Both the top and bottom

*Figure 11 continued on next page*

*Figure 11 continued*

responses are from a held-out condition on which decoders were not trained. (**B**) Actual and predicted modulation of muscle activity when the task is performed using the arm containing that muscle (perf.) or with the other arm (non-perf.). Each thin line plots the modulation for one muscle during one condition (e.g. the deltoid, for top-start forward cycling). Thick lines show medians. The three subpanels correspond to the empirical muscle activity, the train-performing-only decoder (trained only conditions where the task was performed by the arm containing the muscle), and the train-both decoder (trained on conditions of both types). Modulation was always measured for predictions of muscle activity during held-out conditions, for which decoders had to generalize. Data are for monkey E. (**C**) Same data as in B, but plotted in distribution form. Each histogram plots the distribution of empirical (black) or predicted (orange and green) muscle activity across all muscles/conditions. Distributions are plotted separately when the decoded muscle was in the performing (top) or non-performing arm (bottom), to aid comparison within each situation. (**D**) Distribution of arm preference indices for neurons (red), muscles (gray), predictions of a decoder trained only on performing-arm conditions (orange), and predictions of a decoder trained on both performing and non-performing arm conditions (green). Data are for monkey E. Red and gray histograms differ slightly from those in *Figure 4A–B* as they are computed per condition, given that the present analysis focuses on generalization performance for left-out conditions. (**E**) Distribution of decoder weight magnitudes (the contribution of each neuron to decode of muscle activity) for driving-cortex (blue) and non-driving-cortex (red) neurons. Weights are for train-performing-only decoders. Dots show distribution medians. (**F**) As in E, for train-both decoders. (**G–L**) As in A-F, for Monkey F.

DOI: https://doi.org/10.7554/eLife.46159.016

The following figure supplements are available for figure 11:

**Figure supplement 1.** Repetition of analysis in *Figure 11*, analyzing the neural population from the right hemisphere only.
DOI: https://doi.org/10.7554/eLife.46159.017

**Figure supplement 2.** Repetition of analysis in *Figure 11*, analyzing the neural population from the left hemisphere only.
DOI: https://doi.org/10.7554/eLife.46159.018

neurons have random preferences for the two arms (randomly 'mixed' selectivity) orthogonality could be achieved without being explicitly enforced.

Given that subspace-based separation of signals appears to be a common neural strategy, it is worth stressing two points regarding interpretation. First, the weights defining each subspace should not be thought of as mapping directly onto anatomy. For example, a neuron may carry muscle-like signals, and thus contribute to a 'muscle-potent' subspace, even if it is not an output neuron. Such a neuron may drive output neurons, may convey muscle-related signals to areas other than the spinal cord, or may simply be participating in the computations that generate muscle-related signals. In the present study, a neuron can contribute to the 'right-arm' subspace regardless of which hemisphere it resides in. Conversely, an output neuron may contain contributions from both muscle-potent and muscle-null dimensions. This naturally occurs in network models, and is possible so long as non-muscle-like signals average out across output neurons (*Kaufman et al., 2014*). For example, some corticospinal neurons have been shown to have 'mirror' responses to observed actions, despite a lack of corresponding muscle activity (*Kraskov et al., 2014*).

Second, orthogonality of subspaces does not imply that the relevant signals do not interact – indeed it is typically proposed that they do. Preparatory activity is proposed to seed movement-related dynamics (*Afshar et al., 2011*; *Churchland et al., 2006*; *Churchland et al., 2010a*), and movement-period dynamics are proposed to span muscle-null and muscle-potent dimensions (*Churchland et al., 2012*; *Churchland and Shenoy, 2007*). Mixing of signals within a given neuron allows opportunities for such interactions. In the case of the two arms, the opportunity for interaction may be relevant to bimanual coordination.

## Comparison with prior studies of lateralization in M1

Our finding that individual neurons often respond during movements of either arm is in broad agreement with prior primate recording studies. A majority of these studies describe intermixing of right- and left-arm responses in the activity of individual M1 neurons (*Cisek et al., 2003*; *Donchin et al., 2002*; *Donchin et al., 1998*; *Kermadi et al., 1998*; *Steinberg et al., 2002*). However, some studies employing small finger movements report a much smaller percentage of ipsilateral neural responses (*Aizawa et al., 1990*; *Tanji et al., 1988*), suggesting greater lateralization for hand-related versus arm-related computations. Consistent with that interpretation, the hand area of M1 has fewer callosal connections and fewer ipsilateral projections (*Jenny, 1979*; *Jones and Wise, 1977*; *Rouiller et al., 1994*). Yet that interpretation should be considered tentative; fMRI studies find

considerable bilateral tuning in M1 during movement of the digits (*Berlot et al., 2019*; *Diedrichsen et al., 2013*; *Verstynen et al., 2005*).

## Individual-neuron responses are strongly limb-dependent

Two prior studies explicitly compared neural response properties during unimanual movements of each arm. *Cisek et al. (2003)* found that neurons in M1 had limb-dependent preferred directions, yet *Steinberg et al. (2002)* reported preserved preferred directions. At the same time, Steinberg et al. found that left-arm and right-arm reach directions could be independently decoded by separate pools of neurons, suggesting some degree of limb-dependence. The present findings indicate that neural responses are strongly limb-dependent.

Unlike prior studies, our comparisons of response properties concerned their temporal structure, rather than a preferred direction per se, and employ a very different task. Still, our responses are consistent with both *Cisek et al. (2003)* and *Heming et al. (2019)* who found that preferred directions, in response to a load, were independent for the two limbs. Furthermore, although unimanual comparisons were not a central topic of *Donchin et al. (2002)*; *Donchin et al. (1998)*, example units shown in that study do not support the hypothesis of a preferred direction that is consistent across limbs. It thus appears that M1 responses are strongly limb-dependent across a variety of contexts.

Assessed via fMRI, individual voxels within motor cortex have preserved 'finger preference' (pinky, middle or thumb) across limbs (*Diedrichsen et al., 2013*). This might appear in conflict with the present results, where responses were not preserved. A likely resolution is that nearby neurons have responses related to a similar set of joints for both arms, but without preservation of tuning for movement features. For example, a neuron might respond during flexion of the right pinky and extension of the left pinky.

## Possible reasons for ipsilateral motor cortical activity

There exist multiple reasons why motor cortex might be active when the non-driving arm performs the task. A straightforward possibility is that motor cortex employs an abstract limb-independent representation of movement. However, this hypothesis is unlikely given the strongly limb-dependent nature of responses. Alternatively, the two cortices may process different but complementary information. This hypothesis is also unlikely; we found no large signals that were present in the driving cortex but absent in the non-driving cortex.

It is also possible that activity ipsilateral to the moving arm may relate to uncrossed descending connections (*Kuypers, 1981*; *Rosenzweig et al., 2009*). For example, activity in the right motor cortex could exist to drive, via uncrossed connections, muscle activity in the right arm when that arm performs the task. However, prior studies have found little evidence for a robust relationship between M1 and ipsilateral muscle activation. Intracortical microstimulation readily produces contralateral muscle responses (*Kwan et al., 1978*; *Sessle and Wiesendanger, 1982*), yet very rarely generates ipsilateral muscle responses (*Aizawa et al., 1990*). Intracellular recordings of motoneurons reveal no monosynaptic evoked potentials from ipsilateral corticospinal tract stimulation and spike-triggered EMG effects are present only for contralateral muscles (*Soteropoulos et al., 2011*). For these reasons, we suspect that uncrossed projections are unlikely to be the primary reason that the non-driving motor cortex is active.

A further possibility is that activity in the non-driving cortex results from an efference copy of signals generated and employed by the driving cortex. An efference copy could aid coordination between limbs. While this was not necessary in the present task (one hand simply remained still), it may be that efference copy signals are conveyed by default and ignored if they are not needed. Our results are consistent with this possibility, and argue that if it is correct, the efference copy must be quite complete. That is, the driving cortex must convey the majority of the signals it generates, rather than (for example) just the output signals.

Independent of a role in online coordination, an efference copy might allow the non-driving cortex to 'observe' computations in the driving cortex, aiding inter-manual transfer of skills (*Wiestler et al., 2014*). If so, responses in the non-driving cortex might be thought of in much the same way as mirror responses. Mirror neurons have been observed in multiple motor areas (*Fogassi et al., 2005*; *Rizzolatti et al., 1996*), including primary motor cortex (*Dushanova and Donoghue, 2010*; *Vigneswaran et al., 2013*). Even identified pyramidal tract neurons can exhibit

mirror properties (*Kraskov et al., 2014*). These results argue that a common property of neurons in motor areas is a response during movements in which they do not directly participate, possibly for the purposes of learning.

A final possibility is that motor cortical computations are distributed across both hemispheres (*Li et al., 2016*). In the extreme, neurons in the non-driving cortex might simply be viewed the way we view most neurons in the driving cortex; they can contribute to the computation even though they are one or more synapses from the cortico-spinal neurons that will convey the output. This hypothesis is appealing because it could explain the finding that all large signals are shared between hemispheres. More generally, if a randomly chosen neuron from the non-driving cortex has responses that are nearly indistinguishable from a neuron chosen from the driving cortex, perhaps our default assumption should be that they are participating in the same computation. It remains unclear if this hypothesis can be reconciled with the finding that motor cortex inactivation principally affects the contralateral limbs (*Glees and Cole, 1950*; *Liu and Rouiller, 1999*; *Passingham et al., 1983*). If both hemispheres participate in controlling both arms, one would expect a more bilateral deficit. A possible resolution is that the network is sufficiently robust that it can still function when many neurons are inactivated, so long as the output neurons can still convey the necessary commands.

### Ipsilateral arm signals in other brain regions

Cortical areas, subcortical areas, and the spinal cord all contribute to the control of dexterous movements. Indeed, other studies comparing contralateral and ipsilateral movements have found that not only M1, but also the dorsal premotor cortex (*Cisek et al., 2003*; *Kermadi et al., 2000*; *Tanji et al., 1988*), ventral premotor cortex (*Michaels and Scherberger, 2018*), Supplementary Motor Area (*Donchin et al., 2002*; *Gribova et al., 2002*; *Kazennikov et al., 1999*; *Kermadi et al., 2000*; *Kermadi et al., 1998*; *Tanji et al., 1988*), Cingulate Motor Area (*Kermadi et al., 2000*), and the Posterior Parietal Cortex (*Kermadi et al., 2000*) contain neurons which respond to movements of the ipsilateral arm. Furthermore, there are circuits in the brainstem and spinal cord which specifically support the generation of coordinated, rhythmic movements like locomotion (*Duysens and Van de Crommert, 1998*). Other brain regions, such as the Anterior Intraparietal Area, encode movement parameters in a largely limb-independent manner (*Michaels and Scherberger, 2018*), suggesting that the relationship of motor and visuo-motor areas to ipsilateral movements may vary depending on their role in motor computation. In general, movement is generated by a broad, interconnected network of brain and spinal regions. We focused on M1 because it is the cortical region that, based on anatomy and microstimulation results, seemed most likely to have a lateralized representation of movement. Yet, even in M1 we failed to find evidence of strongly lateralized activity.

## Materials and methods

### Behavior

All animal procedures were approved by the Columbia University Institutional Animal Care and Use Committee. Data were collected from two male monkeys (*Macaca mulatta*) while they performed a cycling task for juice reward. Experiments were controlled and data collected under computer control (Real-time Target Machine: Speedgoat, Liebfeld, Switzerland). While performing the task, each monkey sat in a custom primate chair with the head restrained via surgical implant. A video monitor displayed a virtual environment through which the monkey moved. The monkey grasped a custom pedal with each hand. Hands were lightly restrained with fabric tape to keep them in a consistent position on the pedals. The pedal itself was also designed to encourage a consistent hand position, and included a handle and a brace that reduced wrist motion. Pedals turned a crankshaft attached to a motor (Applied Motion Products, Watsonville, CA). A rotary encoder within the motor reported position with 1/8000 cycle precision. Information regarding angular position and its derivatives was used by the real-time system to deliver force commands that produced virtual mass and viscosity. This endowed the pedal with a natural feel, and provided sufficient resistance to ensure robust and repeatable patterns of muscle activity.

Monkeys cycled the pedal to control their position in the virtual environment (*Figure 1A*). The angular position of the pedal was mapped directly to linear position in the virtual world. The task

involved stopping on targets (stationary white squares on the ground) to receive juice reward. On each trial, the monkey progressed from an initial target to a final target. The relationship between world position and pedal position was set such that targets were spaced exactly seven cycles apart. Monkeys performed blocks of twenty consecutive trials. For a given block, one arm was the 'performing arm,' and the other was the 'non-performing arm.' World position was determined by the pedal manipulated by the performing arm. The other pedal was required to remain within a tight position window at the bottom of the cycle (see below). Following completion of a block, the next block was signaled by 5 s of gentle 'buzzing' delivered to one pedal (via the motor) indicating the new performing arm. Blocks were presented in randomized order (*Figure 1C*). Monkeys also executed blocks where both arms cycled together (bimanual task variant), which are not analyzed in this study.

At the start of each trial, the initial target appeared one to two cycles in front of the monkey. The monkey cycled to and 'acquired' this target by stopping on it. The acceptance window was ± 0.15 cycles for Monkey E and ± 0.01 cycles for Monkey F. Holding the target also required remaining below a stringent speed threshold: 0.01 cycles/s for Monkey E and 0.0125 cycles/s for Monkey F. After holding the target for 1000-2000 ms (monkey E) or 600-1000 ms (monkey F) the initial target disappeared and the final target appeared seven cycles ahead of the current position. The monkey then cycled to this final target. To encourage cycling at a swift and consistent rate, we enforced a minimum speed threshold and maximum cycling duration. Once the final target was acquired, the monkey remained still within the target, for 1000-1600 ms, to receive a juice reward. The speed threshold was the same for the initial and final targets. For Monkey F, when the pedal was stopped within the target, the motor provided small stabilizing forces, minimizing the muscle activation needed to hold the pedal stationary.

As a trial was performed, the pedal associated with the non-performing arm had to remain within a window (±0.05 and ±0.07 cycles for monkey E and F) centered at the bottom of the cycle. Movement outside that window caused trial failure followed by a short time-out. Monkeys adopted a stereotyped position within the enforced window and moved little from that position. For the non-performing arm, the position range explored within a trial averaged 0.006 cycles (monkey E) and 0.029 cycles (monkey F).

Within each 20-trial block there were four behavioral conditions (*Figure 1C*). Each condition employed one of two starting pedal positions (top-start and bottom-start), and one of two cycling directions (forward and backward). On top-start conditions, the initial target was located such that the pedal position necessary to acquire that target was at the very top of the cycle. On bottom-start conditions, the necessary pedal position was at the very bottom. In both cases, the final target was exactly seven cycles away, and was thus acquired with the same pedal position. During forward cycling conditions, forward progress in the virtual environment was produced by cycling 'forward', with the arm moving away from the body at the top of the cycle (a motion similar to that of the foot when pedaling a bicycle). During backward cycling conditions, the mapping was reversed. Cycling direction was cued by the color of the landscape in the virtual environment: green for forward, orange for backward. Each of the four combinations of starting position and cycling direction was performed in a sub-block of five trials. Sub-block order was identical for each block (*Figure 1C*).

Monkeys were highly trained on this task before experiments began, and performance was thus consistent across days. For monkey E, data was collected across 19 days of neural recording and 8 days of muscle recording. For monkey F, data was collected across 20 days of neural recording and 6 days of muscle recording.

## Surgical procedures and neural recording

Monkeys were anesthetized and a headcap was implanted under sterile conditions. A 19 mm diameter cylinder (Crist Instruments) was placed above the primary motor cortex of each hemisphere, guided by structural MRI performed prior to surgery. The skull remained intact under the cylinder, covered with a thin layer of dental acrylic. Prior to recording, monkeys were anesthetized and a 3.5 mm diameter burr hole was drilled by hand through the dental acrylic and skull, leaving dura intact. Over the course of the experiment, burr holes were opened at different locations (*Figure 1F,G*). Recordings were made from a variety of depths. Notably, penetrations that entered in anterior M1/posterior PMd could yield recordings in sulcal M1. Following recording, burr holes were closed with dental acrylic, allowing the skull to heal. After the conclusion of our experiment, Monkey E received

a surgery to implant a chronic recording array in the left hemisphere as part of a different study. During this surgery, we confirmed that the burr hole locations lined up with the expected sulcal landmarks.

After opening each burr hole, we first recorded neural activity using conventional single electrodes (Frederick Haer Company). This allowed several days of intracortical microstimulation (ICMS) and muscle palpations to confirm that recordings were within the arm region of M1. For the left hemisphere of Monkey F (*Figure 1G*) minimum ICMS thresholds within the posterior and anterior burr holes were 10 $\mu$A and 20 $\mu$A, respectively. For the right hemisphere, minimum ICMS thresholds within the two posterior burr holes were 10 $\mu$A (medial) and 20 $\mu$A (lateral). These thresholds are consistent with penetrations lying within proximal-arm primary motor cortex, in agreement with the physical location of those four burr holes near the central sulcus. For the more anterior burr hole, the minimum ICMS threshold was 40 $\mu$A, consistent with either posterior dorsal premotor cortex (PMd) or possibly anterior primary motor cortex. A fourth, more posterior burr hole from the right hemisphere was not recorded from due to having no ICMS response up to 150 $\mu$A; this burr hole was presumed to be over S1.

For the left hemisphere of Monkey E (*Figure 1F*) the minimum ICMS thresholds in the posterior and anterior burr holes were 30 $\mu$A and 75 $\mu$A, respectively. These thresholds are consistent with primary motor cortex and PMd, respectively, in agreement with their location. For the right hemisphere of Monkey E, the minimum ICMS thresholds in the posterior and anterior burr holes were 20 and 25 $\mu$A. These thresholds are consistent with primary motor cortex, although the more anterior burr hole likely also contained some recordings from PMd (given its location).

Thus, thresholds and anatomical locations indicate that most recordings were made from primary motor cortex. (Note that locations in *Figure 1F,G* show surface landmarks – penetrations that entered PMd could enter primary motor cortex below). However, a minority of recordings were almost certainly made from premotor cortex. Thus, for both monkeys, key analyses were repeated using data only from the posterior burr-holes. Results were virtually identical, confirming that that properties we report are intrinsic to primary motor cortex, rather than arising from contamination by recordings from premotor cortex.

We recorded neural activity with 24-channel V-Probes (Monkey E), or 32-channel S-Probes (Monkey F) (Plexon Inc, Dallas, TX). We lowered one probe into each hemisphere each day, removing the probe at the end of that session. Probes were moved to different locations within each burr hole on each recording day. Neural signals were processed and recorded using a Digital Hub and 128-channel Neural Signal Processor (Blackrock Microsystems, Salt Lake City, UT). Threshold crossings from each channel were recorded and spike-sorted offline (Plexon Offline Sorter). Unit isolation was assessed based on separation of waveforms in PCA-space, inter-spike interval histograms, and waveform stability over the course of the session. Analyses consider stable, well-isolated single and multi-unit isolations. Multi-unit isolations consisted of identifiable spikes (i.e., not merely threshold crossings) from two (or occasionally more) neurons that could not be distinguished from one another with confidence. All example firing rates shown in figures are from single units. We recorded 263 units in the left hemisphere and 270 units in the right hemisphere for monkey E (8–47 per day), and 338 units in the left hemisphere and 279 units in the right hemisphere for monkey F (12–64 per day).

## EMG recordings

On separate days, interspersed with the neural recordings, we recorded intramuscular EMG signals. We recorded from biceps brachii (long and short head), triceps brachii (medial, long, and lateral heads), deltoid (anterior, lateral, and posterior head), latissimus dorsi, pectoralis, trapezius, and brachioradialis. Pairs of hook-wire electrodes were inserted ~1 cm into the belly of the muscle at the beginning of each session and removed at the end. On each session, 1–3 EMG recordings were made per arm. Electrode voltages were amplified, bandpass filtered (10–500 Hz) and digitized at 1000 Hz (Monkey E) or 30,000 Hz (Monkey F). Recordings were not considered further if they contained significant movement artifact or weak signals. Offline, EMG records were high-pass filtered at 40 Hz, rectified, and smoothed with a 25 ms Gaussian. This produced a measure of intensity versus time, which was then averaged across trials. Trial-averaging of EMG employed the same alignment procedure used when computing trial-averaged firing rates (see below).

## Trial-averaged firing rates

The spike times of each neuron on each trial were converted to a firing-rate by convolving spikes with a 25 ms Gaussian. To produce trial-averaged firing rates, we first aligned all trials on a common time: the moment when the first half-cycle was completed. This nicely aligned behavior across trials during the beginning of each trial. However, because trials lasted multiple seconds, small differences in cycling speed could accumulate and cause considerable misalignment of behavior across trials (*Figure 2A*). The resulting misalignment of single-trial firing rates (*Figure 2B*) would yield an unrepresentative average firing rate (the same problem would impact averages of muscle activity and kinematic variables). Thus, temporal alignment was applied to the middle six cycles, from a half cycle after movement began (i.e. for top-start trials, at the moment when the pedal reached the bottom) until a half-cycle before movement ended.

Alignment was achieved via two steps. First, we temporarily converted firing rates to a position-base rather than a time-base. For example, consider $r_{trial}(t)$, the firing rate of a neuron as a function of time. We constructed $r'_{trial}(\theta)$, where $\theta$ is the angular position of the pedal (from 0 to $12\pi$, as there are six middle cycles). Because all trials now share the same domain, they can be averaged to produce $\overline{r}'(\theta)$. We then converted back to the temporal domain to yield $\overline{r}(t)$. Although there was slight variability across trials and conditions, monkeys cycled very close to 2 Hz. We thus employed the approximation that each cycle lasted 500 ms, and linearly mapped positions $0 - 12\pi$ to times 0-3000 ms (6 cycles with 500 ms per cycle). This mapping ensures that a given time always corresponds to the same position, aiding comparisons across conditions and the pooling of neurons across sessions into a single population. A drawback of this simple mapping is that behavior becomes idealized; that is vertical hand position becomes a perfect sinusoid after alignment. In practice, this drawback was tolerable; vertical hand position was already close to a perfect sinusoid during the middle cycles.

This procedure altered the time-base of individual trials very modestly, yet maintained appropriate alignment across trials (*Figure 2D*), and produced trial-averaged estimates of the firing rate (*Figure 2E*) that are representative of what occurred on single trials. *Figure 2D* plots spike times to illustrate the utility of alignment. However, we stress that temporal alignment was performed after converting spikes to single-trial firing rates via convolution with a Gaussian. Thus, alignment never altered the firing rates themselves, merely the times when those rates were achieved. Alignment was applied only when computing trial-averages, and not in the case of single-trial analyses.

Note that it is not possible to align based on the time-course of hand position unless position is changing. While we could have estimated the very first moment the pedal began to move, and the last moment it was in motion, it was more robust and convenient to simply begin and end at the first and last half-cycles. (Because the first and last half cycles only lasted a few hundred milliseconds, trials remained in reasonable alignment across those times).

Most analyses of firing rates employed the middle four cycles (2-5), excluding the first cycle and the last two cycles. This focused analysis on the steady-state response, rather than on responses associated with starting, stopping, or holding. This aided interpretation in two ways. First, muscle activity in the non-performing arm was particularly weak during middle cycles (in contrast, modest activity was occasionally observed when stopping). Focusing on middle cycles thus largely sidesteps concerns that activity ipsilateral to the performing arm is related to muscle activity in the non-performing arm. Second, we wished to focus key analyses on the rhythmic pattern of firing rate modulation, rather than on overall changes in net firing rate when moving versus not moving. As one example, when predicting muscle activity from neural activity, it is relatively 'easy' to capture the generally elevated activity level during movement, resulting in high $R^2$ values even if predictions fail to capture cycle-by-cycle activity patterns. We wished to avoid this, and to consider predictions successful only if they accounted for rhythmic response aspects.

## Individual-neuron analyses

Analyses were performed based on trial-averaged firing rates, computed as described above, and consider the middle four cycles (cycles 2-5) of the time-varying response. We wished to compare, for each neuron, the strength of modulation when the driven versus non-driven arm performed the task. By modulation, we mean the degree to which a neuron's firing rate varied within cycles, between cycles, and/or between conditions (forward versus backward, and top-start versus bottom-start). For

each neuron, we compiled a single firing rate vector, $r_{driven}$, concatenating the firing rate vectors across the four conditions where the driven arm performed the task. $r_{driven}$ was thus of size $ct$ where $c$ is the number of conditions and $t$ is the number of times during the middle four cycles of one condition. We defined $Modulation_{driven}$ as the standard deviation of $r_{driven}$, which captures the degree to which the average firing rate varies across time and condition. $Modulation_{non-driven}$ was computed analogously, based on $r_{non-driven}$ (the concatenated firing rate across the four conditions where the non-driven arm performed the task).

To assess the degree to which a neuron was more strongly modulated when the driven versus non-driven arm performed the task, we computed an arm preference index:

$$\frac{Modulation_{driven} - Modulation_{non-driven}}{Modulation_{driven} + Modulation_{non-driven}} \tag{1}$$

This arm preference index is zero if a neuron is equally modulated regardless of the arm used, approaches one if modulation is much larger when using the driven arm, and approaches negative one if modulation is much larger when using the non-driven arm.

To compare response patterns when the task was performed with the driven versus non-driven arm, for each neuron we computed Pearson's correlation between $r_{driven}$ and $r_{non-driven}$. $r_{driven}$ and $r_{non-driven}$ were defined as above, with one exception. We wished the present analysis to focus on cycle-by-cycle firing rate patterns during each behavioral condition, rather than in global firing rate changes across conditions. We thus mean-centered the firing rates for each condition (such that the mean was zero) prior to concatenation. This ensured that firing rate patterns could not appear similar due to, for example, strong but otherwise unrelated responses during forward cycling for the two arms.

To determine whether the correlation between $r_{driven}$ and $r_{non-driven}$ was larger than expected by chance, we performed a shuffle control. For each neuron in our population, we replaced its $r_{driven}$ and $r_{non-driven}$ with the $r_{driven}$ from one randomly-selected neuron and the $r_{non-driven}$ from another randomly selected neuron (with replacement). We then computed the average (across the resampled population) Pearson's correlation coefficient between the new $r_{driven}$ and $r_{non-driven}$ values. We repeated this procedure 1000 times. The average correlation of the true population was considered significant if it exceeded 95% of these resampled estimates.

To determine the expected distribution of correlations if neural activity remained similar across two sets of conditions, we compared $r_{top}$ and $r_{bottom}$ for each neuron. $r_{top}$ was constructed by concatenating firing rates for the four conditions (forward and backward, for both arms) with a target location at the top of the cycle. $r_{bottom}$ was constructed analogously for the four conditions with a target location at the bottom of the cycle. Because top-start and bottom-start trials have hand positions 180 degrees out of phase, we used cycles 2-5 when constructing $r_{top}$, and cycles 1.5 – 4.5 when constructing $r_{bottom}$. For each neuron, we computed the Pearson's correlation coefficient between $r_{top}$ and $r_{bottom}$. Because behavior was very similar for top-start and bottom-start conditions (except for the phase difference) these correlations are expected to be near unity unless diluted by sampling error in firing-rate estimates.

## Normalization

Because the absolute voltages of EMG traces are largely arbitrary, the scale of muscle activity could be quite different for different muscles. The response of each muscle was therefore normalized by its range. Neural responses were left un-normalized for individual-neuron analyses. For population-level analyses, responses were normalized to prevent results from being overly biased toward the properties of a few high-rate neurons. For example, principal component analysis (PCA) seeks to capture maximum variance, and a neuron with a firing rate modulation of 100 spikes/s would contribute 25 times as much variance as a neuron with a modulation of 20 spikes/s. Reducing that discrepancy encourages PCA to summarize the response of all neurons. We normalized the firing rate of each neuron using

$$FR_{softnorm} = \frac{FR}{range(FR) + 5} \tag{2}$$

The addition of 5 to the denominator produces 'soft' normalization, and ensures that we don't

magnify the activity of very low-rate neurons. We have used this value previously (*Lara et al., 2018*; *Russo et al., 2018*) as it strikes a reasonable balance between focusing analysis on all neurons while still allowing high firing-rate neurons to contribute somewhat more than very low-rate neurons.

## Firing-rate impact of small movements of the non-performing arm

We wished to control for the possibility that neural responses in the non-driving hemisphere (ipsilateral to the performing arm) might be related to small movements of the (contralateral) non-performing arm. For each trial, we computed the mean (absolute) speed of the non-performing arm. For each condition, we divided trials into those with speeds greater versus less than the median. We did not apply this analysis if there were fewer than eight trials for that condition for that neuron (as sometimes occurred if a neuron was well-isolated for only part of a recording session).

After dividing, we recomputed the mean firing rate for each of the two pools of trials, yielding one firing rate when the non-performing arm moved modestly, and another when it was virtually stationary. For each timepoint, we asked whether these two firing rates were more different than expected given sampling error. This was accomplished via a randomization test in which trials were divided into two groups randomly, rather than based on speed. We performed 1000 such random divisions. Differences were considered significant if they were larger than for 95% of the random divisions.

## Comparison of muscle-activity predictions between hemispheres

To predict muscle activity from neural activity, we used Partial Least Squares (PLS) regression (plsregress in MATLAB). For each set of neurons $X$ and muscles $Y$, PLS regression finds matrices $W, V$, that maximize the covariance between $XW$ and $YV$, under the constraint that $W, V$ are of rank $r$ (which must be specified). PLS is similar to Canonical Correlation Analysis, in that it seeks linear transformations of the data that maximize similarity. However, Canonical Correlation Analysis maximizes correlation, and can focus on small dimensions which may be well-correlated by coincidence. In contrast, PLS regression maximizes covariance and thus seeks correlated signals that are also high variance. Once $W$ is found, $Y$ is predicted from $XW$ via standard linear regression, using the model $XW\hat{B} = Y$. Employing $XW$ (which has only $r$ columns of regressors) rather than $X$ (which has hundreds) greatly reduces overfitting. By setting $B = W\hat{B}$, we obtain a rank $r$ matrix of weights that predicts muscle activity directly from neural activity.

All predictions involved the middle cycles (2-5) of movement. We mean-centered all data, such that average neural and muscle activity was zero for each condition (ensuring that successful prediction involved capturing firing rate structure, not simply an overall mean). We picked one behavioral condition (e.g. top-start, forward, right hand performing) as a test condition. We set the training condition to be the corresponding condition with the opposite starting pedal position (e.g. bottom-start, forward, right hand performing). We ran PLS regression on the training condition to find the rank-$r$ matrix $B$. To select the optimal rank, we selected one cycle from our test condition to serve as validation data. We assessed performance on this validation cycle and selected the rank, $r^*$, and the corresponding weight matrix, $B^*$, that generated the maximal validation $R^2$. We assessed prediction performance (generalization) on the remaining cycles of the test condition. This procedure was repeated for each condition and hemisphere. Generalization performance was quantified based on population percent variance explained:

$$R^2 = 1 - \frac{\left\| Y - Y_{pred} \right\|_f^2}{\left\| Y \right\|_f^2} \tag{3}$$

where $Y_{pred} = XB^*$ and $\|\cdot\|_f$ indicates the Frobenius norm.

## Comparing activity patterns across hemispheres

To assess whether major signals from the driving cortex were absent in the non-driving cortex, we analyzed how well driving cortex activity could be predicted by non-driving cortex activity. For this analysis, we used Principal-Components Regression, which regresses driving cortex activity against the top PC projections of non-driving cortex activity. This enables us to determine both the presence and the relative magnitude of signals within each hemisphere. This analysis was run on the middle

cycles (2-5) of movement. PLS regression yielded similar prediction quality (data not shown). However, PLS regression is specifically formulated to be robust to differences in signal magnitudes between the predictor and target dataset, so it made comparison of signal magnitudes between hemispheres less straightforward.

To predict driving cortex activity from non-driving cortex activity, we performed the following steps. For a given training condition, we ran PCA on $X$, the $t \times n$ (times $\times$ neurons) matrix of firing rates in the non-driving cortex. This yields $W$, the $n \times n$ matrix of projection weights from $X$ into its PC space. Once $W$ is found, the matrix of time-varying firing rates in the driving cortex, $Y$, is predicted from $XW$ via standard linear regression, using the model $XW_r\hat{B} = Y$, where $W_r$ denotes the first $r$ columns of $W$. By setting $B = W_r\hat{B}$, we obtain a rank $r$ matrix of weights that predicts driving cortex activity directly from non-driving cortex activity. We calculated $B$ separately for values of $r$ from 1 to 10. To determine how well $B$ predicts driving cortex activity, we quantified the population $R^2$ (*Equation 3*) on a test condition: the behavior with the same pedaling direction and performing hand, but the opposite starting and stopping position as the training condition.

For comparison, we also calculated how well driving cortex activity could be predicted by other units from the driving cortex. We repeated the steps above, except with $X$ defined as all units except for unit $p$ from the driving cortex, and $Y$ as the activity of unit $p$. Regression weights are thus found for each unit individually, but still restricted to pull from only the top $r$ PC signals calculated from all other units. Generalization performance was calculated using the population $R^2$ (*Equation 3*) on the test condition.

To assess significance, we performed a resampling test to assess the magnitude of difference in $R^2$ expected if there were no underlying division between hemispheres. We randomly re-assigned each neuron to be either a 'left hemisphere' neuron or a 'right hemisphere neuron.' We then calculated the ability of the 'non-driving cortex' to predict the 'driving cortex', versus the ability of the 'driving cortex' to predict itself, as above. This shuffle test was performed 100 times. The difference in $R^2$ between the true driving$\rightarrow$driving and non-driving$\rightarrow$driving predictions was determined to be significant if it exceeded 95/100 of these resampled differences.

To confirm that the above analysis is indeed sufficient to reveal differences between the two hemispheres, we performed two controls. First, we artificially removed a large signal from the non-driving cortex population response, and repeated the analysis. The signal carried by the first PC was removed by reconstructing each neuron's response from all the PCs (which would normally provide a perfect reconstruction) except the first. That is, instead of basing the above analysis on $X$, we based it on $\hat{X} = X\hat{W}\hat{W}^t$, where $\hat{W}$ is the matrix of PCs with the first column missing. This analysis confirmed that, when a signal is missing in the non-driving cortex, the prediction of driving cortex activity from non-driving cortex activity suffers (*Figure 7—figure supplement 1A*). We performed a related control where we altered the size of signals in the non-driving cortex population. To do so, we projected $X$ into its PC space, yielding $X_{PC} = XW$. We divided the first two columns of $X_{PC}$ by two, and multiplied columns three and four by two, yielding $\tilde{X}_{PC}$. We then reconstructed the activity of each neuron: $\tilde{X} = \tilde{X}_{PC}W^t$ and analyzed $\tilde{X}$ as before. This analysis confirmed that prediction suffers if signal sizes are mismatched between the two hemispheres (*Figure 7—figure supplement 1B*).

## Orthogonality of left-arm and right-arm subspaces

We employed PCA to find neural dimensions occupied when the task is performed with a given arm (e.g. the left arm). We then asked how much variance is explained by those dimensions when the task is performed using the other arm (e.g. the right arm). These analyses employed trial-averaged firing rates (as described in this section), and were repeated using estimates of single-trial firing rates (see section describing LFADS below).

Analysis considered triplets of conditions: a training condition, a 'same-arm' test condition and an 'opposite-arm' test condition. The training condition and the same-arm test condition shared the same arm and cycling direction but had different starting positions. The training condition and the opposite-arm test condition were identical but used different arms. Analysis was run eight times per monkey, with every condition serving as a training condition exactly once. Example projections are shown for pairs of test conditions. Quantitative summaries (e.g. variance explained) average across the eight relevant test conditions.

For a given training condition, we applied PCA to the $t \times n$ matrix of firing rates, where $t$ is the number of timepoints during the middle four cycles of movement and $n$ is the number of neurons. This yielded an $n \times n$ matrix of PCs. For each of the two test conditions, we projected the corresponding $t \times n$ matrix of firing rates onto those PCs, and assessed variance explained by each PC.

How close two measured subspaces are to orthogonal depends on the number of neurons in the sample populations. Small sample populations bias estimates away from orthogonality. We therefore explored how our estimates of subspace overlap scaled with neuron count (*Figure 10—figure supplement 1*). These results confirmed that our high neuron-counts (>200 per hemisphere) were important for accurately assessing subspace overlap.

## Fano factor

We employed the Fano factor as a rough measure of whether trial-averaged firing rates are representative of individual-trial responses. The Fano factor is defined as the ratio of the spike-count variance (across trials) to the spike-count mean, with counts made in a temporal window of fixed duration. We computed the Factor for all single-unit isolations with at least ten trials per condition. To compute spike counts, we aligned all trials for a particular condition to the very middle cycle (the time at which the hand position crossed 3.5 cycles, halfway between the beginning and end of the trial). On each trial, spikes were counted in two 250 ms windows: one preceding and one following the alignment time. For each window (treated separately) we computed the mean and variance of the spike counts. We did not modify the time-basis of individual trials. A modified time-base complicates interpretation of spiking variability, and was unnecessary given the relatively short time-spans considered here (behavior drifted little over ± 250 ms). To compute a single Fano factor per dataset, we considered both windows and all neurons and conditions, and used regression to estimate the relationship between spike-count variance and mean (*Churchland et al., 2010a*).

## LFADS

To visualize single-trial neural trajectories, we leveraged a recently-developed machine learning technique: Latent Factor Analysis via Dynamical Systems (LFADS) (*Pandarinath et al., 2018*). LFADS trains a variational auto-encoder to generate single-trial neural firing rates via dynamics instantiated by a recurrent neural network. This approach leverages simultaneous neural recordings to infer latent dynamics that generate neural activity, and leverages those dynamics to generate denoised estimates of single-trial firing rates for each neuron. This allowed us to repeat analyses regarding left-arm and right-arm subspaces (described above for trial-averaged responses) on single-trial firing-rate estimates. We used the LFADS Run Manager (written by Daniel J. O'Shea: https://lfads.github.io/lfads-run-manager/).

An advantage of LFADS is that it can leverage data recorded across sessions. Single-trial latent trajectories are constrained by both the spikes on that trial, and by dynamics inferred across trials and sessions. In the present study, all analyses of single-trial trajectories are performed within a single session, but the inclusion of multiple sessions helps denoise trajectories by better constraining the dynamical model. Sessions were chosen according to two criteria. First, the session had to include a minimum of twenty units (across both hemispheres), all of which remained isolated across the full session (in contrast to trial-averaged data, where we could exclude trials if isolation became poor). Second, we analyzed only sessions where the sampling of response properties was not overly 'lopsided', with one or more conditions having few responsive neurons. This is very unlikely in large datasets, but can occur on single sessions and complicates comparisons of variance captured for different conditions. We therefore required that the neural population variance in the highest-variance behavioral condition not be more than three times the neural population variance in the lowest-variance condition:

$$\sum_n \sigma^2_{n,C+} \leq 3\sum_n \sigma^2_{n,C-} \tag{4}$$

where $\sigma^2_{n,C+}$ denotes the variance of neuron $n$ on the condition with the maximum summed variance and $\sigma^2_{n,C-}$ denotes the variance of neuron $n$ on the condition with the minimum summed variance. For Monkey E, a total of 8 datasets met these criteria, with a range of 21-35 simultaneously recorded

units. For Monkey F, a total of 11 datasets met these criteria, with 20-41 simultaneously recorded units.

LFADS operates upon spike counts, in 1 ms bins, in an equally-sized window per trial. The input is a three-dimensional tensor of size times by neurons by trials. On each trial LFADS infers an initial state and a latent trajectory that evolves according to the learned dynamics. To avoid complexity in the LFADS model, we did not allow it to also infer time-varying inputs. This constrained the model to capture all trial-to-trial differences using different initial states. For this reason, it is important that trials not be overly long. For long trials, trial-to-trial variations that occur late would tend not to be captured unless they can be accounted for via differences in initial state. We specifically wished to capture as much trial-to-trial variability as possible. To allow LFADS the best opportunity to do so, we defined an LFADS 'trial' to be a single cycle of behavior. Thus, each behavioral trial was split into four LFADS trials (one for each cycle from 2 to 5). Each of these spanned a 500 ms window, centered at the time when the hand reached the top pedal location (note that, on different trials, the pedal might travel slightly more or less than one cycle during this window). We trained a stitched multi-day LFADS autoencoder with the following parameters. Bin size: 5 ms; batch size: 32; factor dimensions: 16; generator dimensions: 64; encoder dimensions: 64; learning rate stop: 0.001. Other parameters were left at the default values provided by the LFADS Run Manager.

Analysis of left-arm and right-arm subspaces on single trials paralleled the original analysis based on trial-averages. Single-trial LFADS-inferred firing rates were projected onto left-arm and right-arm subspaces. As in the original analysis, subspaces were found based on a training condition, and the projected data were from two test conditions (same arm but different starting position, or the corresponding condition for the other arm). We computed the variance captured, in each subspace, for each test condition. Although LFADS leveraged data across sessions, the above-analysis steps were performed independently for each session. Subspaces were found based on within-session trial-averages to address the key question of whether subspaces inferred from trial-averages segregate single-trial responses in the same way that they do trial-averaged responses. Similar results were obtained if subspaces were found based on single trials.

## Trajectory tangling

We assessed trajectory tangling as described in *Russo et al. (2018)*. To parallel the other analyses of population activity, trajectory tangling was computed for trajectories during the middle cycles. Trial-averaged neural activity was reduced to the top eight dimensions using PCA. We then calculated tangling, $Q(t)$, at each time point:

$$Q(t) = \max_{t'} \frac{\left\| \dot{x}_t - \dot{x}_{t'} \right\|^2}{\left\| x_t - x_{t'} \right\|^2 + \varepsilon} \tag{5}$$

where $x_t$ is the neural state at time $t$, $\dot{x}_t$ is the temporal derivative of the neural state, $\|\cdot\|$ is the Euclidean norm, and $\varepsilon$ is a small constant that prevents division by zero (here, set to 10% of the total neural variance across the top eight dimensions). Both $t$ and $t'$ index across times within multiple conditions. Thus, $Q(t)$ becomes large if the neural state at time $t$ is close to the neural state at a different time or condition, but the two states have very different derivatives. In the present study, tangling was assessed separately for each hemisphere. Tangling was also assessed separately for the driven and non-driven arms. Thus, $t$ and $t'$ index across all times within four conditions (e.g. the four conditions where the driven arm performs the task). Cumulative distributions of $Q(t)$ were pooled across hemispheres, to create one distribution for the 'driving cortex' and another for the 'non-driving cortex'.

The above analysis was also applied to muscle populations. Tangling was computed for each arm (using only conditions where that arm performed the task) and distributions were then pooled between the two arms to create a single distribution for comparison with neural data.

## Ability of EMG decoders to ignore non-performing-arm-related activity

We assessed whether a linear neural decoder can predict muscle activity in a given arm both when that arm performs the task (and robust EMG needs to be decoded) and when the other arm performs the task (and near-zero EMG should be decoded). As in our previous EMG prediction analysis,

we used PLS regression (described above) and focused on the middle four cycles. For this analysis, we predicted EMG activity using all neurons, regardless of hemisphere.

We first assessed generalization performance using a 'train-moving' decoder, which was trained using only conditions where the relevant arm performed the task, and was then asked to generalize to conditions where the other arm performed the task. This is a potentially challenging form of generalization, as the decoder must predict EMG activity in a situation (arm not moving) very different from the situation in which it was trained (arm moving). We also computed generalization performance of a 'train-both' decoder, trained using a set of conditions that included the relevant arm performing and not performing the task. Generalization was to left-out conditions of each type.

For the train-moving decoder, we used the following division of training, validation, and testing, conditions: *Training*: Direction 1, Start Position 1, Arm moving. *Validation*: Direction 1, Start Position 2, Arm moving (one cycle). *Testing*: Direction 1, Start Position 2, Other arm moving. For the train-both decoder, we used the following division of training, validation, and testing: *Training*: Direction 1, Start Position 1, Arm moving and Direction 1, Start Position 1, Other arm Moving. *Validation*: Direction 1, Start Position 2, Arm moving (one cycle). *Testing*: Direction 1, Start Position 2, Other arm moving.

## Acknowledgements

We thank Y Pavlova for expert animal care. A Russo provided code to calculate trajectory tangling. This work was supported by NINDS 1DP2NS083037, NIH CRCNS R01NS100066, NINDS 1U19NS104649, the Simons Foundation (SCGB#325233 and SCGB#542957), the Grossman Center for the Statistics of Mind, the McKnight Foundation, P30 EY019007, a Klingenstein-Simons Fellowship, and the Searle Scholars Program. KCA was funded by a Simon's Foundation Collaboration on the Global Brain Post-doctoral Fellowship.

## Additional information

### Funding

| Funder | Grant reference number | Author |
| --- | --- | --- |
| National Institute of Neurological Disorders and Stroke | 1DP2NS083037 | Mark M Churchland |
| Simons Foundation | SCGB#325233 | Mark M Churchland |
| McKnight Foundation | P30 EY019007 | Mark M Churchland |
| Klingenstein-Simons Fellowship | | Mark M Churchland |
| Searle Scholars Program | | Mark M Churchland |
| National Institutes of Health | CRCNS R01 NS100066 | Mark M Churchland |
| National Institute of Neurological Disorders and Stroke | 1U19NS104649 | Mark M Churchland |
| Simons Foundation | SCGB#542957 | Mark M Churchland |
| Grossman Center for the Statistics of Mind | | Mark M Churchland |
| Simons Foundation | SCGB Postdoctoral Fellowship | Katherine Cora Ames |

The funders had no role in study design, data collection and interpretation, or the decision to submit the work for publication.

### Author contributions

Katherine Cora Ames, Conceptualization, Data curation, Software, Formal analysis, Investigation, Visualization, Methodology, Writing—original draft, Writing—review and editing; Mark M

Churchland, Conceptualization, Supervision, Funding acquisition, Methodology, Project administration, Writing—review and editing

### Author ORCIDs
Katherine Cora Ames ⬤ https://orcid.org/0000-0003-0657-0162
Mark M Churchland ⬤ http://orcid.org/0000-0001-9123-6526

### Ethics
Animal experimentation: This study was performed in accordance with the recommendations in the Guide for the Care and Use of Laboratory Animals of the National Institutes of Health. All animal protocols were approved by the Columbia University Institutional Animal Care and Use Committee (Protocol number AC-AAAQ7409).

### Decision letter and Author response
Decision letter https://doi.org/10.7554/eLife.46159.026
Author response https://doi.org/10.7554/eLife.46159.027

## Additional files

### Supplementary files
• Transparent reporting form
DOI: https://doi.org/10.7554/eLife.46159.019

### Data availability
Trial-averaged Neural and EMG data, and LFADS-processed single trial data have been uploaded to the Open Science Framework (OSF) and can be accessed at the following link: https://osf.io/859st/.

The following dataset was generated:

| Author(s) | Year | Dataset title | Dataset URL | Database and Identifier |
|---|---|---|---|---|
| Ames K, Churchland M | 2019 | Data for: Motor cortex signals for each arm are mixed across hemispheres and neurons yet partitioned within the population response | https://osf.io/859st/ | Open Science Framework, osf.io/859st |

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
