## [Decision Letter]

Thank you for submitting your article "Motor cortex signals for each arm are mixed across hemispheres and neurons yet partitioned in the population response" for consideration by *eLife*. Your article has been reviewed by three peer reviewers, including Jörn Diedrichsen as the Reviewing Editor and Reviewer #1, and the evaluation has been overseen by Joshua Gold as the Senior Editor. The following individual involved in review of your submission has agreed to reveal their identity: Christian K Machens (Reviewer #3).

The reviewers have discussed the reviews with one another and the Reviewing Editor has drafted this decision to help you prepare a revised submission.

Summary:

The paper by Ames and Churchland provides an important and fresh look into the characteristics of neural activity in motor cortex during ipsilateral arm movements. In many ways the results confirm the study of Donchin et al., 1998, from 20 years ago, showing a high prevalence of ipsilaterally tuned activity and the lack of a tight relation between tuning for contralateral and ipsilateral movement directions. The new analyses, however, reveals the orthogonality of the population activity for contra- and ipsilateral arm, providing important constraints in understanding the function of the widespread ipsilateral activity.

Essential revisions:

The reviewers discussed the points raised in the individual reviews extensively, and agreed on the following major points that should be addressed in the revision.

1) While these findings are quite convincing, the explanation why this bilateral representation is needed (efference copy to support motor planning?) is less clear. For example, finger movements often require an increased coordination effort between both hands, e.g., for bimanual hand actions. This seems at odds with the authors' point that distal (finger) movements are less represented bilaterally than proximal (reach) movements. An additional hypothesis that maybe should be considered in the Discussion is that the non-driving cortex simply "observes" the activity of the driving hemisphere, but does not causally contribute to the driving arm movement. Like in action observation, the neural activity would occur in a muscle null-space (Kraskov et al., 2014). A possible function of such activity (if there is any) may that it helps lay establish networks in the ipsilateral hemisphere, which then underlie the inter-manual transfer of motor skill (Wiestler et al., 2014).

2) How could you confirm that recordings were actually from M1 and not, e.g., from PMd? Did you look for low ICMS thresholds? This is important since bilateral representations in PMd could be quite different from M1.

3) All analyses seem to have been done on trial-averaged data. We think that it is crucial that the authors show whether the orthogonal subspaces still robustly separate the population activity on single trials. It is necessary that single-trial activity also falls into the Null-space for the contralateral hand, such that it can avoid generating contralateral activity. While your analysis shows that on average the activity falls into the null-space, we believe it would be informative to see the spread of the projection. This could be done by projecting single trials onto the orthogonal subspaces, and showing that the information is there, or by applying single-trial dimensionality reduction techniques.

(In this context, it would be interesting to know if single-trial muscle activity in the non-driven arm can be predicted by deviations from the orthogonal subspaces. As far as I understood the limitations of the data, that will not be testable in the current data-set, as muscle activity and brain activity were recorded separately. I don’t ask the authors to do this, but would just like to point out that it could strengthen the authors' hypothesis considerably.)

4) As noted in the Discussion, activity in one M1 hemisphere has little effect on muscle activity in the non-driven arm. Yet, the dimensionality reduction and prediction analyses in Figures 9 and 10 consider the full cross-hemisphere population. To answer the question of why population activity in one hemisphere does not activate the non-driven arm, the dimensionality reduction and prediction analyses should ideally be done on a single hemisphere. The effect may be weaker (because of less data), but it should still be there. Alternatively, the authors could demonstrate that the decoding weights (or weights characterizing the subspaces) properly separate information between the two hemispheres.

5) Even within a hemisphere, an additional concern is that there might be a mix of cell types, with some responsible for direct motor control (e.g., the small proportion of cells with a preference index close to 1), and some responsible for the general computation, say. Previous work in mouse ALM has shown that such a separation can exist, with associated anatomical differences in left/right preference (Li et al., 2015, Nature 519 51-56). I would like the authors (at a minimum) to display some information about the weights of the decoders or subspaces (see also point 2).

6) The authors use PLS to find a low-dimensional linear mapping between population activity X and population activity Y, and evaluated it in generalization setting how well can predict new data. The main results are that muscle activity can be nearly equally well predicted from contra and ipsilateral M1, and that there are hardly any differences in predicting left M1 from left M1 as from right M1. The main limitation of this analysis is that it can show that the two population codes occupy a common linear subspace, but it does not show that the two population codes are structurally the same (or have the same representational geometry – Diedrichsen and Kriegeskorte, 2017). For example, consider the two population depicted in Image 1. Both have a neural dimension that codes for the vertical position of the hand and a neural dimension that codes for the horizontal position of the hand (of course neural dimensions do not cleanly represent specific physical variables, but that's not the point here). This means that you could find a two-dimensional mapping that would predict the population activity in of region A from region B, likely nearly as good as you could predict region A from region A. Furthermore, a specific muscle activity that is a linear combination of position could be read out of the population activity equally well (or bad). However, this obscures the fact that region A overemphasizes the vertical dimension of the movement, whereas region B emphasizes coding for the horizontal dimension.

**Decision letter image 1. desfig1:** Two different neural population codes that occupy the same functional subspace, but that have a different representational geometries.

This problem is especially prevalent as the underlying behavior is relatively simple. While the cycling direction dissociates position, velocity, and muscle activity, the two starting points probably do very little to add new dimensions that the functional subspace that the brain needs to encode. Thus, as long as both hemispheres occupy this relative restricted subspace of neural activity, prediction performance of the PLS model will be quite good. That is, two very different population codes can look very much the same if the experiment does not dissociate the critical dimensions.

We believe that the authors should at least acknowledge these two limitation of their regression approach.

---

## [Author Response]

Essential revisions:The reviewers discussed the points raised in the individual reviews extensively, and agreed on the following major points that should be addressed in the revision.1) While these findings are quite convincing, the explanation why this bilateral representation is needed (efference copy to support motor planning?) is less clear. For example, finger movements often require an increased coordination effort between both hands, e.g., for bimanual hand actions. This seems at odds with the authors' point that distal (finger) movements are less represented bilaterally than proximal (reach) movements. An additional hypothesis that maybe should be considered in the Discussion is that the non-driving cortex simply "observes" the activity of the driving hemisphere, but does not causally contribute to the driving arm movement. Like in action observation, the neural activity would occur in a muscle null-space (Kraskov et al., 2014). A possible function of such activity (if there is any) may that it helps lay establish networks in the ipsilateral hemisphere, which then underlie the inter-manual transfer of motor skill (Wiestler et al., 2014).

We agree with all the above points. First, it is indeed unclear why a bilateral representation is needed, and there remain multiple reasonable possibilities. Second, we agree that there is no clear reason why finger/hand-related responses should be more lateralized (one might expect just the opposite, as the reviewers note).

Further to this point, it is unclear whether finger/hand-related responses are in fact more lateralized. fMRI results reveal bilateral digit-related responses. Finally, we agree that a possible purpose of activity in the non-driving cortex is ‘observation’ of driving-cortex activity for the purpose of inter-manual transfer of skill. We have made the following changes to the manuscript.

1) We have revised the relevant section of the Discussion (Possible reasons for ipsilateral motor cortical activity) to respect the above points. In particular, we now raise the possibility of observation for inter-manual transfer of skill:

“Independent of a role in online coordination, an efference copy might allow the non-driving cortex to ‘observe’ computations in the driving cortex, aiding inter-manual transfer of skills (Wiestler et al., 2014). […] These results argue that a common property of neurons in motor areas is a response during movements in which they do not directly participate, possibly for the purposes of learning.”

2) We have revised the paragraph in the Discussion where we discuss the degree of lateralization found in different studies. We now highlight the lack of clarity regarding whether digit responses are more lateralized.

“Assessed via fMRI, individual voxels within motor cortex have preserved ‘finger preference’ (pinky, middle or thumb) across limbs (Diedrichsen et al., 2013). […] E.g., a neuron might respond during flexion of the right pinky and extension of the left pinky.”

2) How could you confirm that recordings were actually from M1 and not, e.g., from PMd? Did you look for low ICMS thresholds? This is important since bilateral representations in PMd could be quite different from M1.

This is a good question – especially because in one monkey (E) some recordings were almost certainly from caudal PMd (adjacent to M1). Even in monkey F, a few recordings were near the ‘border zone’ (there is of course no clear border). Importantly, we found the same properties regardless of recording location. This is now explicitly stated in the manuscript.

“Results were virtually identical (median arm-preference indices of 0.06 and 0.31) if analysis was restricted to more posterior recordings (excluding the influence of any neurons recorded from PMd or from the border region between M1 and PMd).” [For reference, indices were 0.07 and 0.31 when analyzing all recordings.]

We show in Author response image 1 the relevant figure (Figure 4) replicated with this restricted analysis. In Author response image 2 we show the key analysis (orthogonality of subspaces) after restricting to posterior burr-holes. Results are unchanged. These could be added as supplementary figures if desired.

We have also added to the manuscript details regarding the localization of recordings. This includes new subpanels in Figure 1 showing the location of the burr-holes (keeping in mind that many penetrations traveled well below the illustrated surface features and into the sulcus). We also report ICMS thresholds, which were typically low (10-40 ua, consistent with proximal-arm M1) with the exception of one anterior burr-hole in Monkey E (75 ua, consistent with PMd as noted above). The added text is in the subsection “Surgical Procedures and neural recording”.

**Author response image 1. respfig1:** Figure 4 variant: Neural responses are not strongly lateralized (restricted to neurons from posterior burr holes). (**A**) Histograms of arm preference index for all recorded neurons from posterior burr holes. (**C**) Histograms summarizing, for single neurons, similarity of responses when the task is performed with the driven versus non-driven arm. For each neuron, we computed the correlation between those two responses. *Black* histograms plot the distribution of such correlations across all neurons. *Orange trace*: control demonstrating that high correlations are observed, as expected, when comparing responses during top-start versus bottom-start conditions. *Green trace*: expected distribution if there is no special relationship between responses corresponding to the two arms. This was computed as the correlation between each non-matched pair of neurons.

**Author response image 2. respfig2:** Figure 9 variant: Left arm and right arm related activity lie in orthogonal subspaces. (**A**) Projection of population activity during an example condition performed by the left arm (*blue*) and the same behavior performed by the right arm (*red*), with PCs found using only data during left arm movement. The neural population includes all recorded neurons from both hemispheres, restricted to the posterior burr holes. Analysis considers data from the middle cycles. PCs were found using one behavioral condition, and data shown is from held-out conditions projected into the PC space: either for the condition with the same pedaling direction and hand, but opposite start position, or for the condition with the same start position and pedaling direction, but other hand. Data are for monkey E. (**B**) Cumulative percent variance explained during conditions performed by the left arm (*blue*) and conditions performed by the right arm (*red*), with PCs found using only data during left arm movement. The neural population includes all recorded neurons from both hemispheres, restricted to posterior burr holes. PCs were found using one behavioral condition, and variance explained is assessed on held-out conditions: either the same pedaling direction and performing arm, but opposite start position, or for the same start position and pedaling direction, but opposite performing arm. Each trace is a different generalization condition (four total per arm). Data are for Monkey E. (**C**) Distribution of the absolute value of neuron weights onto the first five “left arm” PCs. Blue traces: units in the right hemisphere (that drive the left arm). Red: units in the left hemisphere (driving the right arm). Histograms are combined across all behavioral conditions, so each neuron contributes 4 conditions x 5 PCs = 20 weights. (G-L) As in A-F, for Monkey F.

3) All analyses seem to have been done on trial-averaged data. We think that it is crucial that the authors show whether the orthogonal subspaces still robustly separate the population activity on single trials. It is necessary that single-trial activity also falls into the Null-space for the contralateral hand, such that it can avoid generating contralateral activity. While your analysis shows that on average the activity falls into the null-space, we believe it would be informative to see the spread of the projection. This could be done by projecting single trials onto the orthogonal subspaces, and showing that the information is there, or by applying single-trial dimensionality reduction techniques.

This is a good suggestion. It is slightly challenging to analyze single trials because we recorded only modest numbers of neurons simultaneously: 8-47 (Monkey E) and 12-64 (Monkey F). This is a consequence of wishing to use v-probes (rather than Utah arrays) to access the central sulcus and to allow flexible recording from a range of sites (yielding higher total neuron-counts that could be achieved with Utah arrays).

Still, on the best days, the number of recorded neurons approached the number of good isolations on a Utah array, making it possible to perform single-trial analyses. We now include single-trial analyses (employing LFADS) to ask whether trial-averages are truly representative. We also address this question by using the Fano Factor to ask whether there is evidence of ‘extra’ variability beyond that expected from roughly Poisson spiking statistics. These analyses are described in a new section (Neural trajectories on single trials) and a new figure (Figure 10). The results demonstrate that single trials behave similarly to trial averages; activity related to the two arms is largely separated into different subspaces. Consistent with this, the Fano Factor is below unity.

(In this context, it would be interesting to know if single-trial muscle activity in the non-driven arm can be predicted by deviations from the orthogonal subspaces. As far as I understood the limitations of the data, that will not be testable in the current data-set, as muscle activity and brain activity were recorded separately. I don’t ask the authors to do this, but would just like to point out that it could strengthen the authors' hypothesis considerably.)

This would indeed be good to know. We presume that the reviewer is correct, and that small single-trial activations of the muscles in the non-performing arm are associated with small excursions into the ‘wrong’ arm’s subspace. Detecting this would of course require simultaneous muscle and neural recordings (which we did not make) and probably also much higher-neuron-count recordings (given that the relevant deviations are likely to be small). Still, this is a clear prediction that may become testable in the future and we agree it should thus be stated. We now do so:

“Our results are consistent with the hypothesis that right-arm-related neural activity falls in dimensions that are muscle-null with respect to the left arm, and vice versa. This hypothesis makes a prediction that could be tested with simultaneous EMG and large-scale neural recordings: small contractions within the non-performing arm should be correlated with small excursions into the corresponding subspace.”

4) As noted in the Discussion, activity in one M1 hemisphere has little effect on muscle activity in the non-driven arm. Yet, the dimensionality reduction and prediction analyses in Figures 9 and 10 consider the full cross-hemisphere population. To answer the question of why population activity in one hemisphere does not activate the non-driven arm, the dimensionality reduction and prediction analyses should ideally be done on a single hemisphere. The effect may be weaker (because of less data), but it should still be there. Alternatively, the authors could demonstrate that the decoding weights (or weights characterizing the subspaces) properly separate information between the two hemispheres.

We agree that it is important to show that what holds for the full population also holds for each hemisphere. This is indeed the case, and is now demonstrated in four added supplementary figures. We added two supplementary figures which reproduce the analysis in Figure 9 for each hemisphere individually (Figure 9—figure supplements 1, 2). We also added two supplementary figures reproducing the results shown in Figure 11 (previously Figure 10) for each hemisphere individually (Figure 11—figure supplements 1, 2). Effects are still very robust. These supplementary figures also allow one to appreciate an additional point: the ability to ignore ‘wrong-arm’ related activity is aided by the fact that not only are subspaces orthogonal, but activity is modestly stronger for the ‘correct’ arm. Thus, the segregation by subspace is particularly clean for the most relevant subspace (e.g., the left-arm subspace for the right cortex, as shown in the top row of Figure 9—figure supplement 1).

We like these added analyses, but have kept the analysis in the main figures focused on the full population for both practical and conceptual reasons. Practically, it allows for fewer subpanels (e.g., Figure 9 can have 12 subpanels, rather than 24). Conceptually, at this point we want to shift the conversation towards thinking about motor cortex as a full population. The results in the earlier figures establish that both sides respond during movement of both arms, and contain similar signals. Anatomically, the two sides are heavily interconnected. It thus becomes reasonable to ask, ‘if all the relevant signals are shared across a large bi-lateral population, how can arm-specific signals become separated.’ Indeed, it is a particularly stringent and interesting test to ask whether the signals related to the two arms can be separated *without* needing to know which side each neuron was recorded from. We have modified the beginning of the relevant section to reflect this motivation.

We also agree that it is worth adding the subspace / decoding weights to the relevant figures (now 9 and 11). This has been done. As expected, both hemispheres contribute to both spaces/decode (with the expected small driving-cortex bias).

5) Even within a hemisphere, an additional concern is that there might be a mix of cell types, with some responsible for direct motor control (e.g., the small proportion of cells with a preference index close to 1), and some responsible for the general computation, say. Previous work in mouse ALM has shown that such a separation can exist, with associated anatomical differences in left/right preference (Li et al., 2015, Nature 519 51-56). I would like the authors (at a minimum) to display some information about the weights of the decoders or subspaces (see also point 2).

We agree it is interesting to consider whether there might be sub-populations which correspond more tightly to particular parameters of the computation. We now show distributions of projection weights for PCA in Figure 9, and EMG decoding weights in Figure 11. These distributions don’t support the hypothesis of a distinct subpopulation with large weights for a given space (e.g., a subpopulation of ‘pure left-arm’ neurons). There is a somewhat continuous distribution. Still, we would not want to entirely rule out that hypothesis. For example, it seems plausible that cortico-motoneurons could be more ‘pure’, and would thus occupy the tail of the distribution.

6) The authors use PLS to find a low-dimensional linear mapping between population activity X and population activity Y, and evaluated it in generalization setting how well can predict new data. The main results are that muscle activity can be nearly equally well predicted from contra and ipsilateral M1, and that there are hardly any differences in predicting left M1 from left M1 as from right M1. The main limitation of this analysis is that it can show that the two population codes occupy a common linear subspace, but it does not show that the two population codes are structurally the same (or have the same representational geometry – Diedrichsen and Kriegeskorte, 2017).For example, consider the two population depicted in Image 1. Both have a neural dimension that codes for the vertical position of the hand and a neural dimension that codes for the horizontal position of the hand (of course neural dimensions do not cleanly represent specific physical variables, but that's not the point here). This means that you could find a two-dimensional mapping that would predict the population activity in of region A from region B, likely nearly as good as you could predict region A from region A. Furthermore, a specific muscle activity that is a linear combination of position could be read out of the population activity equally well (or bad). However, this obscures the fact that region A overemphasizes the vertical dimension of the movement, whereas region B emphasizes coding for the horizontal dimension.This problem is especially prevalent as the underlying behavior is relatively simple. While the cycling direction dissociates position, velocity, and muscle activity, the two starting points probably do very little to add new dimensions that the functional subspace that the brain needs to encode. Thus, as long as both hemispheres occupy this relative restricted subspace of neural activity, prediction performance of the PLS model will be quite good. That is, two very different population codes can look very much the same if the experiment does not dissociate the critical dimensions.We believe that the authors should at least acknowledge these two limitation of their regression approach.

These are excellent points. We agree that the PLS approach has two notable limitations. The first is that PLS would not distinguish between two population responses that are ‘stretched’ relative to one another (i.e., share the same signals but to quite different degrees). The second is that a simple task might provide limited ability to ‘see’ dimensions that are different. For example, a more complex task might evoke a 50-dimensional response, with the first ten dimensions shared between the driving and non-driving hemispheres, and rest being completely different. Thus, we need to be careful to stress that our conclusion is simply that the signals carried by the dominant dimensions are very similar. We have made the following revisions:

1) We have altered our analysis to address the first limitation. We now employ PCA-regression rather than PLS regression. To appreciate why PCA-regression addresses this limitation, consider a scenario of the type described by the reviewers: signals X1 and Χ2 are present in both hemispheres, but X1 is emphasized in the driving cortex and Χ2 is emphasized in the non-driving cortex. If we apply PCA to the driving cortex population, and project onto the first PC, we will recover X1. When we regress driving-cortex activity against X1, it will capture considerable variance (say 60%). If we apply PCA to the non-driving cortex population, and project onto the first PC, we will recover Χ2. When we regress the driving-cortex population against Χ2, it will capture rather less variance (say 30%). Thus, PCA regression, unlike PLS regression, will reveal a mismatch in signal sizes. We therefore now employ PCA regression for this analysis (other analyses still use PLS regression). We also confirm (Figure 7—figure supplement 1) that this analysis would indeed reveal differences if they were there. The relevant figure panels (Figure 7E-G) have been updated, as has the relevant text. (We also continue to note, for completeness, that PLS regression yielded similar results.)

2) We have added the following text to explicitly acknowledge the second limitation:

“Thus, the dominant signals – those present in the top handful of principal components – are nearly identical between the driving and non-driving cortices. A caveat is that smaller signals could be quite different, which would be difficult to discern given our relatively simple task. Activity spans a modest number of dimensions when cycling at steady-state (5 principal components account for ~80% of the variance). It is quite possible that differences in smaller signals, carried by other dimensions, could be uncovered with added task complexity (e.g., cycling at different speeds or under different loads).”